# Rethinking the Capacity of Graph Neural Networks for Branching Strategy

**Ziang Chen**
Massachusetts Institute of Technology
ziang@mit.edu

**Jialin Liu**
University of Central Florida
jialin.liu@ucf.edu

**Xiaohan Chen**      **Xinshang Wang**      **Wotao Yin**
Alibaba US, DAMO Academy
{xiaohan.chen,xinshang.w,wotao.yin}@alibaba-inc.com

## Abstract

Graph neural networks (GNNs) have been widely used to predict properties and heuristics of mixed-integer linear programs (MILPs) and hence accelerate MILP solvers. This paper investigates the capacity of GNNs to represent strong branching (SB), the most effective yet computationally expensive heuristic employed in the branch-and-bound algorithm. In the literature, message-passing GNN (MP-GNN), as the simplest GNN structure, is frequently used as a fast approximation of SB and we find that not all MILPs's SB can be represented with MP-GNN. We precisely define a class of "MP-tractable" MILPs for which MP-GNNs can accurately approximate SB scores. Particularly, we establish a universal approximation theorem: for any data distribution over the MP-tractable class, there always exists an MP-GNN that can approximate the SB score with arbitrarily high accuracy and arbitrarily high probability, which lays a theoretical foundation of the existing works on imitating SB with MP-GNN. For MILPs without the MP-tractability, unfortunately, a similar result is impossible, which can be illustrated by two MILP instances with different SB scores that cannot be distinguished by any MP-GNN, regardless of the number of parameters. Recognizing this, we explore another GNN structure called the second-order folklore GNN (2-FGNN) that overcomes this limitation, and the aforementioned universal approximation theorem can be extended to the entire MILP space using 2-FGNN, regardless of the MP-tractability. A small-scale numerical experiment is conducted to directly validate our theoretical findings.

## 1 Introduction

Mixed-integer linear programming (MILP) involves optimization problems with linear objectives and constraints, where some variables must be integers. These problems appear in various fields, from logistics and supply chain management to planning and scheduling, and are in general NP-hard. The branch-and-bound (BnB) algorithm [33] is the core of a MILP solver. It works by repeatedly solving relaxed versions of the problem, called linear relaxations, which allow the integer variables to take on fractional values. If a relaxation's solution satisfies the integer requirements, it is a valid solution to the original problem. Otherwise, the algorithm divides the problem into two subproblems and solves their relaxations. This process continues until it finds the best solution that meets all the constraints.

**Branching** is the process of dividing a linear relaxation into two subproblems. When branching, the solver selects a variable with a fractional value in the relaxation's solution and create two new subproblems. In one subproblem, the variable is forced to be less than or equal to the nearest integer

below the fractional value. In the other, it is bounded above the fractional value. The branching variable choice is critical because it can impact the solver's efficiency by orders of magnitude.

A well-chosen branching variable can lead to a significant improvement in the lower bound, which is a quantity that can quickly prove that a subproblem and its further subdivisions are infeasible or not promising, thus reducing the total number of subproblems to explore. This means fewer linear relaxations to solve and faster convergence to the optimal solution. On the contrary, a poor choice may result in branches that do little to improve the bounds or reduce the solution space, thus leading to a large number of subproblems to be solved, significantly increasing the total solution time. The choice of which variable to branch on is a pivotal decision. This is where **branching strategies**, such as strong branching and learning to branch, come into play, evaluating the impact of different branching choices before making a decision.

**Strong branching (SB)** [3] is a sophisticated strategy to select the *most promising branches* to explore. In SB, before actually performing a branch, the solver tentatively branches on several variables and calculates the potential impact of each branch on the objective function. This "look-ahead" strategy evaluates the quality of branching choices by solving linear relaxations of the subproblems created by the branching. The variable that leads to the most significant improvement in the objective function is selected for the actual branching. Usually recognized as the most effective branching strategy, *SB often results in a significantly lower number of subproblems to resolve during the branch-and-bound (BnB)* process compared to other methods [18]. As such, SB is frequently utilized directly or as a fundamental component in cutting-edge solvers.

While SB can significantly reduce the size of the BnB search space, it comes with *high computational cost*: evaluating multiple potential branches at each decision point requires solving many LPs. This leads to a *trade-off* between the time spent on SB and the overall time saved due to a smaller search space. In practice, MILP solvers use heuristics to limit the use of SB to where it is most beneficial.

**Learning to branch (L2B)** introduces a new approach by incorporating machine learning (ML) to develop branching strategies, offering new solutions to address this trade-off. This line of research begins with imitation learning [2, 5, 19, 24, 25, 29, 35, 56, 58], where models, including SVM, decision tree, and neural networks, are trained to mimic SB outcomes based on the features of the underlying MILP. They aim to *create a computationally efficient strategy that achieves the effectiveness of SB on specific datasets*. Furthermore, in recent reinforcement learning approaches, mimicking SB continues to take crucial roles in initialization or regularization [45, 60].

While using a heuristic (an ML model) to approximate another heuristic (the SB procedure) may seem counterintuitive, it is important to recognize the potential benefits. The former can significantly reduce the time required to make branching decisions as effectively as the latter. As MILPs become larger and more complex, the computational cost of SB grows at least cubically, but some ML models grow quadratically, even just linearly after training on a set of similar MILPs. Although SB can theoretically solve LP relaxations in parallel, the time required for different LPs may vary greatly, and there is a lack of GPU-friendly methods that can effectively utilize starting bases for warm starts. In contrast, ML models, particularly GNNs, are more amenable to efficient implementation on GPUs, making them a more practical choice for accelerating the branching variable selection process. Furthermore, additional problem-specific characteristics can be incorporated into the ML model, allowing it to make more informed branching decisions tailored to each problem instance.

**Graph neural network (GNN)** stands out as an effective class of ML models for L2B, surpassing other models like SVM and MLP, due to the excellent scalability and the permutation-invariant/equivariant property. To utilize a GNN on a MILP, one first conceptualizes the MILP as a graph and the GNN is then applied to that graph and returns a branching decision. This approach [15, 19] has gained prominence in not only L2B but various other MILP-related learning tasks [13, 17, 26, 30, 32, 36, 40, 43, 44, 48, 50–52, 54, 57]. More details are provided in Section 2.

Despite the widespread use of GNNs on MILPs, a theoretical understanding remains largely elusive. A vital concept for any ML model, including GNNs, is its **capacity** or **expressive power** [27, 34, 46], which in our context is their ability to accurately approximate the mapping from MILPs to their SB results. Specifically, this paper aims to answer the following question:

> *Given a distribution of MILPs, is there a GNN model capable of mapping each MILP problem to its strong branching result with a specified level of precision?* (1.1)

**Related works and our contributions.** While the capacity of GNNs for general graph tasks, such as node and link prediction or function approximation on graphs, has been extensively studied [4, 10, 22, 28, 37, 39, 47, 55, 59], their capacities in approximating SB remains largely unexplored. The closest studies [11, 12] have explored GNNs' ability to represent properties of linear programs (LPs) and MILPs, such as feasibility, boundedness, or optimal solutions, but have not specifically focused on branching strategies. Recognizing this gap, our paper makes the following contributions:

- In the context of L2B using GNNs, we first focus on the most widely used type: message-passing GNNs (MP-GNNs). Our study reveals that MP-GNNs can reliably predict SB results, but only for a specific class of MILPs that we introduce as *message-passing-tractable* (MP-tractable). We prove that for any distribution of MP-tractable MILPs, there exists an MP-GNN capable of accurately predicting their SB results. This finding establishes a theoretical basis for the widespread use of MP-GNNs to approximate SB results in current research.

- Through a counter-example, we demonstrate that MP-GNNs are *incapable* of predicting SB results beyond the class of MP-tractable MILPs. The counter-example consists of two MILPs with distinct SB results to which all MP-GNNs, however, yield identical branching predictions.

- For general MILPs, we explore the capabilities of *second-order folklore GNNs (2-FGNNs)*, a type of higher-order GNN with enhanced expressive power. Our results show that 2-FGNNs can reliably answer question (1.1) positively, effectively replicating SB results across any distribution of MILP problems, surpassing the capabilities of standard MP-GNNs.

Overall, as a series of works have empirically shown that learning an MP-GNN as a fast approximation of SB significantly benefits the performance of an MILP solver on specific data sets [2, 5, 19, 24, 25, 29, 35, 56, 58], our goal is to determine whether there is room, in theory, to further understand and improve the GNNs' performance on this task.

## 2 Preliminaries and problem setup

We consider the MILP defined in its general form as follows:

$$\min_{x \in \mathbb{R}^n} c^\top x, \quad \text{s.t. } Ax \circ b, \ \ell \le x \le u, \ x_j \in \mathbb{Z}, \ \forall j \in I, \tag{2.1}$$

where $A \in \mathbb{R}^{m \times n}$, $b \in \mathbb{R}^m$, $c \in \mathbb{R}^n$, $\circ \in \{\le, =, \ge\}^m$ is the type of constraints, $\ell \in (\{-\infty\} \cup \mathbb{R})^n$ and $u \in (\mathbb{R} \cup \{\infty\})^n$ are the lower bounds and upper bounds of the variable $x$, and $I \subset \{1, 2, \ldots, n\}$ identifies which variables are constrained to be integers.

**Graph Representation of MILP.** Here we present an approach to represent MILP as a bipartite graph, termed the *MILP-graph*. This conceptualization was initially proposed by [19] and has quickly become a prevalent model in ML for MILP-related tasks. The MILP-graph is defined as a tuple $G = (V, W, A, F_V, F_W)$, where the components are specified as follows: $V = \{1, 2, \ldots, m\}$ and $W = \{1, 2, \ldots, n\}$ are sets of nodes representing the constraints and variables, respectively. An edge $(i, j)$ connects node $i \in V$ to node $j \in W$ if the corresponding entry $A_{ij}$ in the coefficient matrix of (2.1) is non-zero, with $A_{ij}$ serving as the edge weight. $F_V$ are features/attributes of constraints, with features $v_i = (b_i, \circ_i)$ attached to node $i \in V$. $F_W$ are features/attributes of variables, with features $w_j = (c_j, \ell_j, u_j, \delta_I(j))$ attached to node $j \in W$, where $\delta_I(j) \in \{0, 1\}$ indicates whether the variable $x_j$ is integer-constrained.

We define $\mathcal{N}_W(i) =: \{j \in W : A_{ij} \neq 0\} \subset W$ as the neighbors of $i \in V$ and similarly define $\mathcal{N}_V(j) =: \{i \in V : A_{ij} \neq 0\} \subset V$. This graphical representation *completely* describes a MILP's information, allowing us to interchangeably refer to a MILP and its graph throughout this paper. An illustrative example is presented in Figure 1. We also introduce a space of MILP-graphs:

**Definition 2.1** (Space of MILP-graphs)**.** *We use $\mathcal{G}_{m,n}$ to denote the collection of all MILP-graphs induced from MILPs of the form* (2.1) *with $n$ variables and $m$ constraints.*[1]

**Message-passing graph neural networks (MP-GNNs)** are a class of GNNs that operate on graph-structured data, by passing messages between nodes in a graph to aggregate information from their local neighborhoods. In our context, the input is an aforementioned MILP-graph $G = (V, W, A, F_V, F_W)$, and each node in $W$ is associated with a real-number output. We use the standard MP-GNNs for MILPs in the literature [12, 19].

Specifically, the initial layer assigns features $s_i^0, t_j^0$ for each node as

---

[1] Rigorously, the space $\mathcal{G}_{m,n} \cong \mathbb{R}^{m \times n} \times \mathbb{R}^n \times \mathbb{R}^m \times (\mathbb{R} \cup \{-\infty\})^n \times (\mathbb{R} \cup \{+\infty\})^n \times \{\le, =, \ge\}^m \times \{0, 1\}^n$ is equipped with product topology, where all Euclidean spaces have standard Eudlidean topologies, discrete spaces $\{-\infty\}$, $\{+\infty\}$, $\{\le, =, \ge\}$, and $\{0, 1\}$ have the discrete topologies, and all unions are disjoint unions.

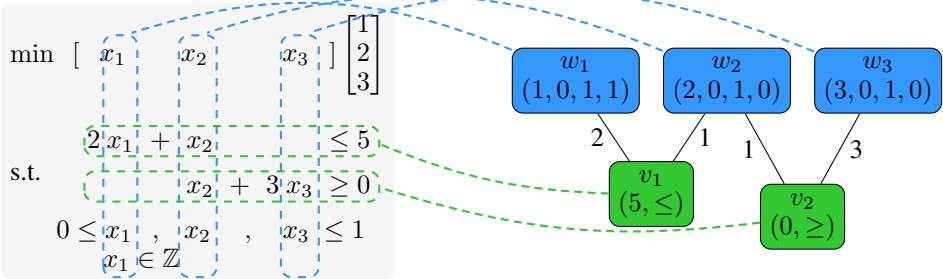

Figure 1: An illustrative example of MILP and its graph representation.

- $s_i^0 = p^0(v_i)$ for each constraint $i \in V$, and $t_j^0 = q^0(w_j)$ for each variable $j \in W$.

Then message-passing layers $l = 1, 2, \ldots, L$ update the features via

- $s_i^l = p^l\big(s_i^{l-1}, \sum_{j \in \mathcal{N}_W(i)} f^l(t_j^{l-1}, A_{ij})\big)$ for each constraint $i \in V$, and

- $t_j^l = q^l\big(t_j^{l-1}, \sum_{i \in \mathcal{N}_V(j)} g^l(s_i^{l-1}, A_{ij})\big)$ for each variable $j \in W$.

Finally, the output layer produces a read-number output $y_j$ for each node $j \in W$:

- $y_j = r\big(\sum_{i \in V} s_i^L, \sum_{j \in W} t_j^L, t_j^L\big)$.

In practice, functions $\{p^l, q^l, f^l, g^l\}_{l=1}^L, r, p^0, q^0$ are learnable and usually parameterized with multi-linear perceptrons (MLPs). In our theoretical analysis, we assume for simplicity that those functions are continuous on given domains. The space of MP-GNNs is introduced as follows.

**Definition 2.2** (Space of MP-GNNs). *We use $\mathcal{F}_{\text{MP-GNN}}$ to denote the collection of all MP-GNNs constructed as above with $p^l, q^l, f^l, g^l, r$ being continuous with $f^l(\cdot, 0) \equiv 0$ and $g^l(\cdot, 0) \equiv 0$.[2]*

Overall, any MP-GNN $F \in \mathcal{F}_{\text{MP-GNN}}$ maps a MILP-graph $G$ to a $n$-dim vector: $y = F(G) \in \mathbb{R}^n$.

**Second-order folklore graph neural networks (2-FGNNs)** are an extension of MP-GNNs designed to overcome some of the capacity limitations. It is proved in [55] the expressive power of MP-GNNs can be measured by the Weisfeiler-Lehman test (WL test [53]). To enhance the ability to identify more complex graph patterns, [42] developed high-order GNNs, inspired by high-order WL tests [9]. Since then, there has been growing literature about high-order GNNs and other variants including high-order folklore GNNs [4, 20–22, 38, 61]. Instead of operating on individual nodes of the given graph, 2-FGNNs operate on *pairs of nodes* (regardless of whether two nodes in the pair are neighbors or not) and the neighbors of those pairs. We say two node pairs are neighbors if they share a common node. Let $G = (V, W, A, F_V, F_W)$ be the input graph. The initial layer performs:

- $s_{ij}^0 = p^0(v_i, w_j, A_{ij})$ for each constraint $i \in V$ and each variable $j \in W$, and

- $t_{j_1 j_2}^0 = q^0(w_{j_1}, w_{j_2}, \delta_{j_1 j_2})$ for variables $j_1, j_2 \in W$,

where $\delta_{j_1 j_2} = 1$ if $j_1 = j_2$ and $\delta_{j_1 j_2} = 0$ otherwise. For internal layers $l = 1, 2, \ldots, L$, compute

- $s_{ij}^l = p^l\big(s_{ij}^{l-1}, \sum_{j_1 \in W} f^l(t_{j_1 j}^{l-1}, s_{ij_1}^{l-1})\big)$ for all $i \in V, j \in W$, and

- $t_{j_1 j_2}^l = q^l\big(t_{j_1 j_2}^{l-1}, \sum_{i \in V} g^l(s_{ij_2}^{l-1}, s_{ij_1}^{l-1})\big)$ for all $j_1, j_2 \in W$.

The final layer produces the output $y_j$ for each node $j \in W$:

- $y_j = r\big(\sum_{i \in V} s_{ij}^L, \sum_{j_1 \in W} t_{j_1 j}^L\big)$.

Similar to MP-GNNs, the functions within 2-FGNNs, including $\{p^l, q^l, f^l, g^l\}_{l=1}^L, r, p^0, q^0$, are also learnable and typically parameterized with MLPs. The space of 2-FGNNs is defined with:

**Definition 2.3.** *We use $\mathcal{F}_{\text{2-FGNN}}$ to denote the set of all 2-FGNNs with continuous $p^l, q^l, f^l, g^l, r$.*

Any 2-FGNN, $F \in \mathcal{F}_{\text{2-FGNN}}$, maps a MILP-graph $G$ to a $n$-dim vector: $y = F(G)$. While MP-GNNs and 2-FGNNs share the same input-output structure, their internal structures differ, leading to distinct expressive powers.

---

[2]We require $f^l, g^l$ yield 0 when the edge weight is 0 to avoid the discontinuity of functions in $\mathcal{F}_{\text{MP-GNN}}$.

# 3 Imitating strong branching by GNNs

In this section, we present some observations and mathematical concepts underlying the imitation of strong branching by GNNs. This line of research, which aims to replicate SB strategies through GNNs, has shown promising empirical results across a spectrum of studies [19, 24, 25, 35, 48, 56, 58], yet it still lacks theoretical foundations. Its motivation stems from two key observations introduced earlier in Section 1, which we elaborate on here in detail.

**Observation I:** SB is notably effective in reducing the size of the BnB search space. This size is measured by the size of the BnB tree. Here, a "tree" refers to a hierarchical structure of "nodes", each representing a decision point or a subdivision of the problem. The tree's size corresponds to the number of these nodes. For instance, consider the instance "neos-3761878-oglio" from MIPLIB [23]. When solved using SCIP [7, 8] under standard configurations, the BnB tree size is $851$, and it takes $61.04$ seconds to attain optimality. However, disabling SB, along with all branching rules dependent on SB, results in an increased BnB tree size to $35548$ and an increased runtime to $531.0$ seconds.

**Observation II:** SB itself is computationally expensive. In the above experiment under standard settings, SB consumes an average of $70.40\%$ of the total runtime, $42.97$ out of $61.04$ seconds in total.

Therefore, there is a clear need of approximating SB with efficient ML models. Ideally, if we can substantially reduce the SB calculation time from $42.97$ seconds to a negligible duration while maintaining its effectiveness, the remaining runtime of $61.04 - 42.97 = 18.07$ seconds would become significantly more efficient.

To move forward, we introduce some basic concepts related to SB.

**Concepts for SB.** SB begins by identifying candidate variables for branching, typically those with non-integer values in the solution to the linear relaxation but which are required to be integers. Each candidate is then assigned a *SB score*, a non-negative real number determined by creating two linear relaxations and calculating the objective improvement. A higher SB score indicates the variable has a higher priority to be chosen for branching. Variables that do not qualify as branching candidates are assigned a zero score. Compiling these scores for each variable results in an $n$-dimensional SB score vector, denoted as $\mathrm{SB}(G) = (\mathrm{SB}(G)_1, \mathrm{SB}(G)_2, \ldots, \mathrm{SB}(G)_n)$.

Consequently, the task of approximating SB with GNNs can be described with a mathematical language: *finding an $F \in \mathcal{F}_{\text{MP-GNN}}$ or $F \in \mathcal{F}_{\text{2-FGNN}}$ such that $F(G) \approx \mathrm{SB}(G)$.* Formally, it is:

**Formal statement of Problem** (1.1)**:** *Given a distribution of $G$, is there $F \in \mathcal{F}_{\text{MP-GNN}}$ or $F \in \mathcal{F}_{\text{2-FGNN}}$ such that $\|F(G) - \mathrm{SB}(G)\|$ is smaller than some error tolerance with high probability?*

To provide clarity, we present a formal definition of SB scores:

**Definition 3.1** (LP relaxation with a single bound change)**.** *Pick a $G \in \mathcal{G}_{m,n}$. For any $j \in \{1, 2, \ldots, n\}$, $\hat{l}_j \in \{-\infty\} \cup \mathbb{R}$, and $\hat{u}_j \in \mathbb{R} \cup \{+\infty\}$, we denote by $\mathrm{LP}(G, j, \hat{l}_j, \hat{u}_j)$ the following LP problem obtained by changing the lower/upper bound of $x_j$ in the LP relaxation of (2.1):*

$$\min_{x \in \mathbb{R}^n} \ c^\top x, \quad \text{s.t. } Ax \circ b, \ \hat{l}_j \leq x_j \leq \hat{u}_j, \ l_{j'} \leq x_{j'} \leq u_{j'} \text{ for } j' \in \{1, 2, \ldots, n\} \backslash \{j\}.$$

**Definition 3.2** (Strong branching scores)**.** *Let $G \in \mathcal{G}_{m,n}$ be a MILP-graph associated with the problem (2.1) whose LP relaxation is feasible and bounded. Denote $f_{\text{LP}}^*(G) \in \mathbb{R}$ as the optimal objective value of the LP relaxation of $G$ and denote $x_{\text{LP}}^*(G) \in \mathbb{R}^n$ as the optimal solution with the smallest $\ell_2$-norm. The SB score $\mathrm{SB}(G)_j$ for variable $x_j$ is defined via*

$$\mathrm{SB}(G)_j = \begin{cases} 0, & \text{if } j \notin I, \\ (f_{\text{LP}}^*(G, j, l_j, \hat{u}_j) - f_{\text{LP}}^*(G)) \cdot (f_{\text{LP}}^*(G, j, \hat{l}_j, u_j) - f_{\text{LP}}^*(G)), & \text{otherwise}, \end{cases}$$

*where $f_{\text{LP}}^*(G, j, l_j, \hat{u}_j)$ and $f_{\text{LP}}^*(G, j, \hat{l}_j, u_j)$ are the optimal objective values of $\mathrm{LP}(G, j, l_j, \hat{u}_j)$ and $\mathrm{LP}(G, j, \hat{l}_j, u_j)$ respectively, with $\hat{u}_j = \lfloor x_{\text{LP}}^*(G)_j \rfloor$ being the largest integer no greater than $x_{\text{LP}}^*(G)_j$ and $\hat{l}_j = \lceil x_{\text{LP}}^*(G)_j \rceil$ being the smallest integer no less than $x_{\text{LP}}^*(G)_j$, for $j = 1, 2, \ldots, n$.*

**Remark: LP solution with the smallest $\ell_2$-norm.** We only define the SB score for MILP problems with feasible and bounded LP relaxations; otherwise the optimal solution $x_{\text{LP}}^*(G)$ does not exist. If the LP relaxation of $G$ admits multiple optimal solutions, then the strong branching score $\mathrm{SB}(G)$ depends on the choice of the particular optimal solution. To guarantee that the SB score is uniquely defined, in Definition 3.2, we use the optimal solution with the smallest $\ell_2$-norm, which is unique.

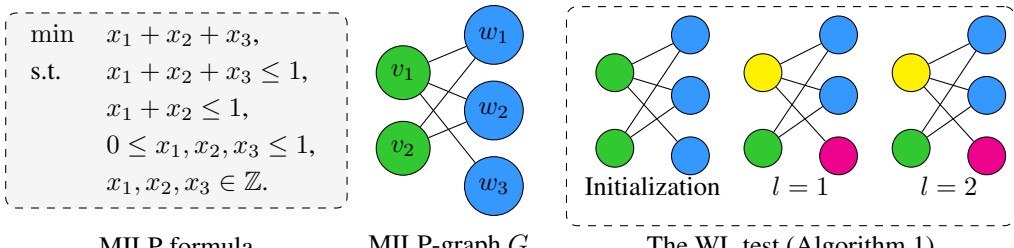

$$
\begin{aligned}
\min \quad & x_1 + x_2 + x_3, \\
\text{s.t.} \quad & x_1 + x_2 + x_3 \le 1, \\
& x_1 + x_2 \le 1, \\
& 0 \le x_1, x_2, x_3 \le 1, \\
& x_1, x_2, x_3 \in \mathbb{Z}.
\end{aligned}
$$

MILP formula      MILP-graph $G$      The WL test (Algorithm 1)

Figure 2: An illustrative example of color refinement and partitions. Initially, all variables share a common color due to their identical node attributes, as do the constraint nodes. After a round of the WL test, $x_1$ and $x_2$ retain their shared color, while $x_3$ is assigned a distinct color, as it connects solely to the first constraint, unlike $x_1$ and $x_2$. Similarly, the colors of the two constraints can also be differentiated. Finally, this partition stabilizes, resulting in $\mathcal{I} = \{\{1\}, \{2\}\}$, $\mathcal{J} = \{\{1, 2\}, \{3\}\}$.

**Remark: SB at leaf nodes.** While the strong branching score discussed here primarily pertains to root SB, it is equally relevant to SB at leaf nodes within the BnB framework. By interpreting the MILP-graph $G$ in Definition 3.2 as representing the subproblems encountered during the BnB process, we can extend our findings to strong branching decisions at any point in the BnB tree. Here, *root SB* refers to the initial branching decisions made at the root of the BnB tree, while *leaf nodes* represent subsequent branching points deeper in the tree, where similar SB strategies can be applied.

**Remark: Other types of SB scores.** Although this paper primarily focuses on the product SB scores (where the SB score is defined as the product of objective value changes when branching up and down), our analysis can extend to other forms of SB scores in [14]. (Refer to Appendix D.1)

## 4 Main results

### 4.1 MP-GNNs can represent SB for MP-tractable MILPs

In this subsection, we define a class of MILPs, named message-passing-tractable (MP-tractable) MILPs, and then show that MP-GNNs can represent SB within this class.

To define MP-tractability, we first present the Weisfeiler-Lehman (WL) test [53], a well-known criterion for assessing the expressive power of MP-GNNs [55]. The WL test in the context of MILP-graphs is stated in Algorithm 1. It follows exactly the same updating rule as the MP-GNN, differing only in the local updates performed via hash functions.

---
**Algorithm 1** The WL test for MILP-Graphs

---
**Require:** A graph instance $G \in \mathcal{G}_{m,n}$ and iteration limit $L > 0$.
1: Initialize with $C_0^V(i) = \text{HASH}_0^V(v_i)$, $C_0^W(j) = \text{HASH}_0^W(w_j)$.
2: **for** $l = 1, 2, \cdots, L$ **do**
3:     $C_l^V(i) = \text{HASH}_l^V \left( C_{l-1}^V(i), \left\{ \left\{ \left( C_{l-1}^W(j), A_{ij} \right) : j \in \mathcal{N}_W(i) \right\} \right\} \right)$.
4:     $C_l^W(j) = \text{HASH}_l^W \left( C_{l-1}^W(j), \left\{ \left\{ \left( C_{l-1}^W(i), A_{ij} \right) : i \in \mathcal{N}_V(j) \right\} \right\} \right)$.
5: **end for**
6: **Output:** Final colors $C_L^V(i)$ for all $i \in V$ and $C_L^W(j)$ for all $j \in V$.

---

The WL test can be understood as a **color refinement algorithm**. In particular, each vertex in $G$ is initially assigned a color $C_0^V(i)$ or $C_0^W(j)$ according to its initial feature $v_i$ or $w_j$. Then the vertex colors $C_l^V(i)$ and $C_l^W(j)$ are iteratively refined via aggregation of neighbors' information and corresponding edge weights. If there is no collision of hash functions[3], then two vertices are of the same color at some iteration if and only if at the previous iteration, they have the same color and the same multiset of neighbors' information and corresponding edge weights. Such a color refinement process is illustrated by an example shown in Figure 2.

One can also view a vertex coloring as a **partition**, i.e., all vertices are partitioned into several classes such that two vertices are in the same class if and only if they are of the same color. After each round

---
[3]Here, "no collision of a hash function" indicates that the hash function doesn't map two distinct inputs to the same output during the WL test on a specific instance. Another stronger assumption, commonly used in WL test analysis [27], assumes that all hash functions are injective.

of Algorithm 1, the partition always becomes finer if no collision happens, though it may not be strictly finer. The following theorem suggests that this partition eventually stabilizes or converges, with the final limit uniquely determined by the graph $G$, independent of the hash functions selected.

**Theorem 4.1** ([11, Theorem A.2]). *For any $G \in \mathcal{G}_{m,n}$, the vertex partition induced by Algorithm 1 (if no collision) will converge within $\mathcal{O}(m+n)$ iterations to a partition $(\mathcal{I}, \mathcal{J})$, where $\mathcal{I} = \{I_1, I_2, \ldots, I_s\}$ is a partition of $\{1, 2, \ldots, m\}$ and $\mathcal{J} = \{J_1, J_2, \ldots, J_t\}$ is a partition of $\{1, 2, \ldots, n\}$, and that partition $(\mathcal{I}, \mathcal{J})$ is uniquely determined by the input graph $G$.*

With the concepts of color refinement and partition, we can introduce the core concept of this paper:

**Definition 4.2** (Message-passing-tractability). *For $G \in \mathcal{G}_{m,n}$, let $(\mathcal{I}, \mathcal{J})$ be the partition as in Theorem 4.1. We say that $G$ is message-passing-tractable (MP-tractable) if for any $p \in \{1, 2, \ldots, s\}$ and $q \in \{1, 2, \ldots, t\}$, all entries of the submatrix $(A_{ij})_{i \in I_p, j \in J_q}$ are the same. We use $\mathcal{G}_{m,n}^{\mathrm{MP}} \subset \mathcal{G}_{m,n}$ to denote the subset of all MILP-graphs in $\mathcal{G}_{m,n}$ that are MP-tractable.*

In order to help readers better understand the concept of "MP-tractable", let's examine the MILP instance shown in Figure 2. After numerous rounds of WL tests, the partition stabilizes to $\mathcal{I} = \{\{1\}, \{2\}\}$ and $\mathcal{J} = \{\{1, 2\}, \{3\}\}$. According to Definition 4.2, one must examine the following submatrices to determine whether the MILP is MP-tractable:

$$A[1, 1:2] = [1, 1], \quad A[2, 1:2] = [1, 1], \quad A[1, 3] = [1], \quad A[2, 3] = [0].$$

All elements within each submatrix are identical. Hence, this MILP is indeed MP-tractable. To rigorously state our result, we require the following assumption of the MILP data distribution.

**Assumption 4.3.** $\mathbb{P}$ *is a Borel regular probability measure on $\mathcal{G}_{m,n}$ and $\mathbb{P}[\mathrm{SB}(G) \in \mathbb{R}^n] = 1$.*

Borel regularity is a "minimal" assumption that is actually satisfied by almost all practically used data distributions such as normal distributions, discrete distributions, etc. Let us also comment on the other assumption $\mathbb{P}[\mathrm{SB}(G) \in \mathbb{R}^n] = 1$. In Definition 3.2, the linear relaxation of $G$ is feasible and bounded, which implies $f_{\mathrm{LP}}^*(G) \in \mathbb{R}$. However, it is possible for a linear program that is initially bounded and feasible to become infeasible upon adjusting a single variable's bounds, potentially resulting in $f_{\mathrm{LP}}^*(G, j, l_j, \hat{u}_j) = +\infty$ or $f_{\mathrm{LP}}^*(G, j, \hat{l}_j, u_j) = +\infty$ and leading to an infinite SB score: $\mathrm{SB}(G)_j = +\infty$. Although we ignore such a case by assuming $\mathbb{P}[\mathrm{SB}(G) \in \mathbb{R}^n] = 1$, it is straightforward to extend all our results by simply representing $+\infty$ as $-1$ considering $\mathrm{SB}(G)_j$ as a non-negative real number, thus avoiding any collisions in definitions.

Based on the above assumptions, as well as an extra assumption: $G$ is message-passing tractable with probability one, we can show the existence of an MP-GNN capable of accurately mapping a MILP-graph $G$ to its corresponding SB score, with an arbitrarily high degree of precision and probability. The formal theorem is stated as follows.

**Theorem 4.4.** *Let $\mathbb{P}$ be any probability distribution over $\mathcal{G}_{m,n}$ that satisfies Assumption 4.3 and $\mathbb{P}[G \in \mathcal{G}_{m,n}^{\mathrm{MP}}] = 1$. Then for any $\varepsilon, \delta > 0$, there exists a GNN $F \in \mathcal{F}_{\mathrm{MP\text{-}GNN}}$ such that*

$$\mathbb{P}[\|F(G) - \mathrm{SB}(G)\| \leq \delta] \geq 1 - \epsilon.$$

The proof of Theorem 4.4 is deferred to Appendix A, with key ideas outlined here. First, we show that if Algorithm 1 produces identical results for two MP-tractable MILPs, they must share the same SB score. That is, if two MP-tractable MILPs have different SB scores, the WL test (or equivalently MP-GNNs) can capture this distinction. Building on this result, along with a generalized version of the Stone-Weierstrass theorem and Luzin's theorem, we reach the final conclusion.

Let us compare our findings with [12] that establishes the existence of an MP-GNN capable of directly mapping $G$ to one of its optimal solutions, under the assumption that $G$ must be **unfoldable**. Unfoldability means that, after enough rounds of the WL test, each node receives a distinct color assignment. Essentially, it assumes that the WL test can differentiate between all nodes in $G$, and the elements within the corresponding partition $(\mathcal{I}, \mathcal{J})$ have cardinality one: $|I_p| = 1$ and $|J_q| = 1$ for all $p \in \{1, 2, \ldots, s\}$ and $q \in \{1, 2, \ldots, t\}$. Consequently, any unfoldable MILP must be MP-tractable because the submatrices under the partition of an unfoldable MILP $(A_{ij})_{i \in I_p, j \in J_q}$ must be $1 \times 1$ and obviously satisfy the condition in Definition 4.2. However, the reverse assertion is not true: The example in Figure 2 serves as a case in point—it is MP-tractable but not unfoldable. Therefore, unfoldability is a stronger assumption than MP-tractability. Our Theorem 4.4 demonstrates that, *to illustrate the expressive power of MP-GNNs in approximating SB, MP-tractability suffices; we do not need to make assumptions as strong as those required when considering MP-GNN for approximating the optimal solution.*

## 4.2 MP-GNNs cannot universally represent SB beyond MP-tractability

Our next main result is that MP-GNNs do not have sufficient capacity to represent SB scores on the entire MILP space without the assumption of MP-tractability, stated as follows.

**Theorem 4.5.** *There exist two MILP problems with different SB scores, such that any MP-GNN has the same output on them, regardless of the number of parameters.*

There are infinitely many pairs of examples proving Theorem 4.5, and we show two simple examples:

$$\begin{aligned} \min \quad & x_1 + x_2 + x_3 + x_4 + x_5 + x_6 + x_7 + x_8, \\ \text{s.t.} \quad & x_1 + x_2 \geq 1, \ x_2 + x_3 \geq 1, \ x_3 + x_4 \geq 1, \ x_4 + x_5 \geq 1, \ x_5 + x_6 \geq 1, \\ & x_6 + x_7 \geq 1, \ x_7 + x_8 \geq 1, \ x_8 + x_1 \geq 1, \ 0 \leq x_j \leq 1, \ x_j \in \mathbb{Z}, \ 1 \leq j \leq 8, \end{aligned} \tag{4.1}$$

$$\begin{aligned} \min \quad & x_1 + x_2 + x_3 + x_4 + x_5 + x_6 + x_7 + x_8, \\ \text{s.t.} \quad & x_1 + x_2 \geq 1, \ x_2 + x_3 \geq 1, \ x_3 + x_1 \geq 1, \ x_4 + x_5 \geq 1, \ x_5 + x_6 \geq 1, \\ & x_6 + x_4 \geq 1, \ x_7 + x_8 \geq 1, \ x_8 + x_7 \geq 1, \ 0 \leq x_j \leq 1, \ x_j \in \mathbb{Z}, \ 1 \leq j \leq 8. \end{aligned} \tag{4.2}$$

We will prove in Appendix B that these two MILP instances have different SB scores, but they cannot be distinguished by any MP-GNN in the sense that for any $F \in \mathcal{F}_{\text{MP-GNN}}$, inputs (4.1) and (4.2) lead to the same output. Therefore, it is impossible to train an MP-GNN to approximate the SB score meeting a required level of accuracy with high probability, independent of the complexity of the MP-GNN. Any MP-GNN that accurately predicts one MILP's SB score will necessarily fail on the other. We also remark that our analysis for (4.1) and (4.2) can be generalized easily to any aggregation mechanism of neighbors' information when constructing the MP-GNNs, not limited to the sum aggregation as in Section 2.

The MILP instances on which MP-GNNs fail to approximate SB scores, (4.1) and (4.2), are not MP-tractable. It can be verified that for both (4.1) and (4.2), the partition as in Theorem 4.1 is given by $\mathcal{I} = \{I_1\}$ with $I_1 = \{1, 2, \ldots, 8\}$ and $\mathcal{J} = \{J_1\}$ with $J_1 = \{1, 2, \ldots, 8\}$, i.e., all vertices in $V$ form a class and all vertices in $W$ form the other class. Then the matrices $(A_{ij})_{i \in I_1, j \in J_1}$ and $(\bar{A}_{ij})_{i \in I_1, j \in J_1}$ are just $A$ and $\bar{A}$, the coefficient matrices in (4.1) and (4.2), and have both 0 and 1 as entries, which does not satisfies Definition 4.2.

Based on Theorem 4.5, we can directly derive the following corollary by considering a simple discrete uniform distribution $\mathbb{P}$ on only two instances (4.1) and (4.2).

**Corollary 4.6.** *There exists a probability distribution $\mathbb{P}$ over $\mathcal{G}_{m,n}$ satisfying Assumption 4.3 and constants $\epsilon, \delta > 0$, such that for any MP-GNN $F \in \mathcal{F}_{\text{MP-GNN}}$, it holds that*

$$\mathbb{P}[\|F(G) - \text{SB}(G)\| \geq \delta] \geq \epsilon.$$

This corollary indicates that the assumption of MP-tractability in Theorem 4.4 is not removable.

## 4.3 2-FGNNs are capable of universally representing SB

Although the universal approximation of MP-GNNs for SB scores is conditioned on the MP-tractability, we find an unconditional positive result stating that when we increase the order of GNNs a bit, it is possible to represent SB scores of MILPs, regardless of the MP-tractability.

**Theorem 4.7.** *Let $\mathbb{P}$ be any probability distribution over $\mathcal{G}_{m,n}$ that satisfies Assumption 4.3. Then for any $\varepsilon, \delta > 0$, there exists a GNN $F \in \mathcal{F}_{\text{2-FGNN}}$ such that*

$$\mathbb{P}[\|F(G) - \text{SB}(G)\| \leq \delta] \geq 1 - \epsilon.$$

The proof of Theorem 4.7 leverages the second-order folklore Weisfeiler-Lehman (2-FWL) test. We show that for any two MILPs, whether MP-tractable or not, identical 2-FWL results imply they share the same SB score, thus removing the need for MP-tractability. Details are provided in Appendix C.

Theorem 4.7 establishes the existence of a 2-FGNN that can approximate the SB scores of MILPs well with high probability. This is a fundamental result illustrating the possibility of training a GNN to predict branching strategies for MILPs that are not MP-tractable. In particular, for any probability distribution $\mathbb{P}$ as in Corollary 4.6 on which MP-GNNs fail to predict the SB scores well, Theorem 4.7 confirms the capability of 2-FGNNs to work on it.

However, it's worth noting that 2-FGNNs typically have higher computational costs, both during training and inference stages, compared to MP-GNNs. This computational burden comes from

the fact that calculations of 2-FGNNs reply on pairs of nodes instead of nodes, as we discussed in Section 2. To mitigate such computational challenges, one could explore the use of sparse or local variants of high-order GNNs that enjoy cheaper information aggregation with strictly stronger separation power than GNNs associated with the original high-order WL test [41].

## 4.4 Practical insights of our theoretical results

Theorem 4.4 and Corollary 4.6 indicate the significance of MP-tractability in practice. Before attempting to train a MP-GNN to imitate SB, practitioners can first verify if the MILPs in their dataset satisfy MP-tractability. If the dataset contains a substantial number of MP-intractable instances, careful consideration of this approach is necessary, and 2-FGNNs may be more suitable according to Theorem 4.7. Notably, assessing MP-tractability relies solely on conducting the WL test (Algorithm 1). This algorithm is well-established in graph theory and benefits from abundant resources and repositories for implementation. Moreover, it operates with polynomial complexity (detailed below), which is reasonable compared to solving MILPs.

**Complexity of verifying MP-tractability.** To verify the MP-tractability of a MILP, one requires at most $\mathcal{O}(m+n)$ color refinement iterations according to Theorem 4.1. The complexity of each iteration is bounded by the number of edges in the graph [49]. In our context, it is bounded by the number of nonzeros in matrix $A$: $\text{nnz}(A)$. Therefore, the overall complexity is $\mathcal{O}((m+n) \cdot \text{nnz}(A))$, which is linear in terms of $(m+n)$ and $\text{nnz}(A)$. In contrast, solving a MILP or even calculating its all the SB scores requires significantly higher complexity. To calculate the SB score of each MILP, one needs to solve at most $n$ LPs. We denote the complexity of solving each LP as $\text{CompLP}(m,n)$. Therefore, the overall complexity of calculating SB scores is $\mathcal{O}(n \cdot \text{CompLP}(m,n))$. Note that, currently, there is still no strongly polynomial-time algorithm for LP, thus this complexity is significantly higher than that of verifying MP-tractability.

While verifying MP-tractability is polynomial in complexity, the complexity of GNNs is still not guaranteed. Theorems 4.4 and 4.7 address existence, not complexity. In other words, this paper answers the question of *whether GNNs can represent the SB score*. To explore *how well GNNs can represent SB*, further investigation is needed.

**Frequency of MP-tractability.** In practice, the occurrence of MP-tractable instances is highly dependent on the dataset. In both Examples 4.1 and 4.2 (both MP-intractable), all variables exhibit symmetry, as they are assigned the same color by the WL test, which fails to distinguish them. Conversely, in the 3-variable example in Figure 2 (MP-tractable), only two of the three variables, $x_1$ and $x_2$, are symmetric. Generally, the frequency of MP-tractability depends on **the level of symmetry** in the data — higher levels of symmetry increase the risk of MP-intractability. This phenomenon is commonly seen in practical MILP datasets, such as MIPLIB 2017 [23]. According to [12], approximately one-quarter of examples show significant symmetry in over half of the variables.

## 5 Numerical results

We implement numerical experiments to validate our theoretical findings in Section 4.

**Experimental settings**: We train an MP-GNN and a 2-FGNN with $L = 2$, where we replace the functions $f^l(t_j^{l-1}, A_{ij})$ and $g^l(s_i^{l-1}, A_{ij})$ in the MP-GNN by $A_{ij}f^l(t_j^{l-1})$ and $A_{ij}g^l(s_i^{l-1})$ to guarantee that they take the value 0 whenever $A_{ij} = 0$. For both GNNs, $p^0, q^0$ are parameterized as linear transformations followed by a non-linear activation function; $\{p^l, q^l, f^l, g^l\}_{l=1}^{L}$ are parameterized as 3-layer multi-layer perceptrons (MLPs) with respective learnable parameters; and the output mapping $r$ is parameterized as a 2-layer MLP. All layers map their input to a 1024-dimensional vector and use the ReLU activation function. With $\theta$ denoting the set of all learnable parameters of a network, we train both MP-GNN and 2-FGNN to fit the SB scores of the MILP dataset $\mathcal{G}$, by minimizing $\frac{1}{2}\sum_{G \in \mathcal{G}} \|F_\theta(G) - \text{SB}(G)\|^2$ with respect to $\theta$, using Adam [31]. The networks and training scheme is implemented with Python and TensorFlow [1]. The numerical experiments are conducted on a single NVIDIA Tesla V100 GPU for two datasets:

- We randomly generate 100 MILP instances, with 6 constraints and 20 variables, that are *MP-tractable* with probability 1. SB scores are collected using SCIP [6]. More details about instance generation are provided in Appendix E.

- We train the MP-GNN and 2-FGNN to fit the SB scores of (4.1) and (4.2), i.e., the dataset only consists of two instances that are *not MP-tractable*.

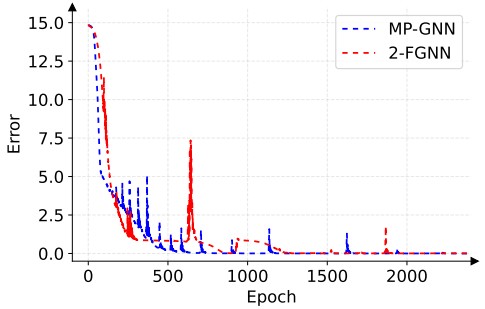 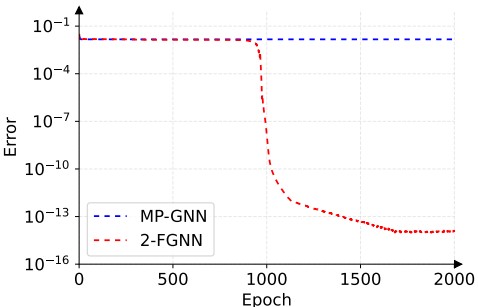

(a) MP-tractable MILPs: Both MP-GNN and 2-FGNN can fit the SB scores.

(b) MP-intractable MILPs (4.1) and (4.2): 2-FGNN can fit SB scores while MP-GNN can not.

Figure 3: Numerical results of MP-GNN and 2-FGNN for SB score fitting. In the right figure, the training error of MP-GNN on MP-intractable examples does not decrease after however many epochs.

**Experimental results**: The numerical results are displayed in Figure 3. One can see from Figure 3a that both MP-GNN and 2-FGNN can approximate the SB scores over the dataset of random MILP instances very well, which validates Theorem 4.4 and Theorem 4.7. As illustrated in Figure 3b, 2-FGNN can perfectly fit the SB scores of (4.1) and (4.2) simultaneously while MP-GNN can not, which is consistent with Theorem 4.5 and Theorem 4.7 and serves as a numerical verification of the capacity differences between MP-GNN and 2-FGNN for SB prediction. The detailed exploration of training and performance evaluations of GNNs is deferred to future work to maintain a focused investigation on the theoretical capabilities of GNNs in this paper.

**Number of parameters:** In Figure 3b, the behavior of MP-GNN remains unchanged regardless of the number of parameters used, as guaranteed by Theorem 4.5. This error is intrinsically due to the structure of MP-intractable MILPs and cannot be reduced by adding parameters. Conversely, 2-FGNN can achieve near-zero loss with sufficient parameters, as guaranteed by Theorem 4.7 and confirmed by our numerical experiments. To further verify this, we tested 2-FGNN with embedding sizes from 64 to 2,048. All models reached near-zero errors, though epoch counts varied, as shown in Table 1. The results suggest that larger embeddings improve model capacity to fit counterexamples. The gains level off beyond an embedding size of 1,024 due to increased training complexity.

Table 1: Epochs required to reach specified errors with varying embedding sizes for 2-FGNN.

| Embedding size | 64 | 128 | 256 | 512 | 1,024 | 2,048 |
|---|---|---|---|---|---|---|
| **Epochs to reach $10^{-6}$ error** | 16,570 | 5,414 | 2,736 | 1,442 | 980 | 1,126 |
| **Epochs to reach $10^{-12}$ error** | 18,762 | 7,474 | 4,412 | 2,484 | 1,128 | 1,174 |

**Larger instances:** While our study primarily focuses on theory and numerous empirical studies have shown the effectiveness of GNNs in branching strategies (as noted in Section 1), we conducted experiments on larger instances to further assess the scalability of this approach. We trained an MP-GNN on 100 large-scale set covering problems, each with 1,000 variables and 2,000 constraints, generated following the methodology in [19]. The MP-GNN achieved a training loss of $1.94 \times 10^{-4}$, calculated as the average $\ell_2$ norm of errors across all training instances.

## 6 Conclusion

In this work, we study the expressive power of two types of GNNs for representing SB scores. We find that MP-GNNs can accurately predict SB results for MILPs within a specific class termed "message-passing-tractable" (MP-tractable). However, their performance is limited outside this class. In contrast, 2-FGNNs, which update node-pair features instead of node features as in MP-GNNs, can universally approximate the SB scores on every MILP dataset or for every MILP distribution. These findings offer insights into the suitability of different GNN architectures for varying MILP datasets, particularly considering the ease of assessing MP-tractability. We also comment on limitations and future research topics. Although the universal approximation result is established for MP-GNNs and 2-FGNNs to represent SB scores, it is still unclear what is the required complexity/number of parameters to achieve a given precision. It would thus be interesting and more practically useful to derive some quantitative results. In addition, exploring efficient training strategies or alternatives of higher order GNNs for MILP tasks is an interesting and significant future direction.

## Acknowledgments and Disclosure of Funding

We would like to express our deepest gratitude to Prof. Pan Li from the School of Electrical and Computer Engineering at Georgia Institute of Technology (GaTech ECE), for insightful discussions on second-order folklore GNNs and their capacities for general graph tasks. We would also like to thank Haoyu Wang from GaTech ECE for helpful discussions during his internship at Alibaba US DAMO Academy.

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

# A Proof of Theorem 4.4

This section presents the proof of Theorem 4.4. We define the separation power of WL test in Definition A.1 and prove that two MP-tractable MILP-graphs, or two vertices in a single MP-tractable graph, indistinguishable by WL test must share the same SB score in Theorem A.3. In other words, WL test has sufficient separation power to distinguish MP-tractable MILP graphs, or vertices in a single MP-tractable graph, with different SB scores.

Before stating the major result, we first introduce some definitions and useful theorems.

**Definition A.1.** *Let $G, \bar{G} \in \mathcal{G}_{m,n}$ and let $C_l^V(i), C_l^W(j)$ and $\bar{C}_l^V(i), \bar{C}_l^W(j)$ be the colors generated by the WL test (Algorithm 1) for $G$ and $\bar{G}$. We say $G \overset{W}{\sim} \bar{G}$ if $\{\{C_L^V(i) : i \in V\}\} = \{\{\bar{C}_L^V(i) : i \in V\}\}$ and $C_L^W(j) = \bar{C}_L^W(j), \forall j \in W$ holds for any $L$ and any hash functions.*

**Theorem A.2** ([11, Theorem A.2]). *The partition defined in Theorem 4.1 satisfies:*

(a) $v_i = v_{i'}, \forall i, i' \in I_p, p \in \{1, 2, \ldots, s\}$,

(b) $w_j = w_{j'}, \forall j, j' \in J_q, q \in \{1, 2, \ldots, t\}$,

(c) $\{\{A_{ij} : j \in J_q\}\} = \{\{A_{i'j} : j \in J_q\}\}, \forall i, i' \in I_p, p \in \{1, 2, \ldots, s\}, q \in \{1, 2, \ldots, t\}$,

(d) $\{\{A_{ij} : i \in I_p\}\} = \{\{A_{ij'} : i \in I_p\}\}, \forall j, j' \in J_q, p \in \{1, 2, \ldots, s\}, q \in \{1, 2, \ldots, t\}$,

*where $\{\{\}\}$ denotes the multiset considering both the elements and the multiplicities.*

In Theorem A.2, conditions (a) and (b) mean vertices in the same class share the same features, while conditions (c) and (d) state that vertices in the same class interact with another class with the same multiset of weights. In other words, for any $p \in \{1, 2, \ldots, s\}$ and $q \in \{1, 2, \ldots, t\}$, different rows/columns of the submatrix $(A_{ij})_{i \in I_p, j \in J_q}$ provide the same multiset of entries.

With the above preparations, we can state and prove the main result now.

**Theorem A.3.** *For any $G, \bar{G} \in \mathcal{G}_{m,n}^{\mathrm{MP}}$ with $\mathrm{SB}(G) \in \mathbb{R}^n$ and $\mathrm{SB}(\bar{G}) \in \mathbb{R}^n$, the followings are true:*

(a) *If $G \overset{W}{\sim} \bar{G}$, then $\mathrm{SB}(G) = \mathrm{SB}(\bar{G})$.*

(b) *If $C_L^W(j_1) = C_L^W(j_2)$ holds for any $L$ and any hash functions, then $\mathrm{SB}(G)_{j_1} = \mathrm{SB}(G)_{j_2}$.*

*Proof.* (a) Since $G \overset{W}{\sim} \bar{G}$, after applying some permutation on $V$ (relabelling vertices in $V$) in the graph $\bar{G}$, the two $G$ and $\bar{G}$ share the same partition $\mathcal{I} = \{I_1, I_2, \ldots, I_s\}$ and $\mathcal{J} = \{J_1, J_2, \ldots, J_t\}$ as in Theorem A.2 and we have

- For any $p \in \{1, 2, \ldots, s\}$, $v_i = \bar{v}_i$ is constant over all $i \in I_p$,

- For any $q \in \{1, 2, \ldots, t\}$, $w_j = \bar{w}_j$ is constant over all $j \in J_q$,

- For any $p \in \{1, 2, \ldots, s\}$ and $q \in \{1, 2, \ldots, t\}$, $\{\{A_{ij} : j \in J_q\}\} = \{\{\bar{A}_{ij} : j \in J_q\}\}$ is constant over all $i \in I_p$,

- For any $p \in \{1, 2, \ldots, s\}$ and $q \in \{1, 2, \ldots, t\}$, $\{\{A_{ij} : i \in I_p\}\} = \{\{\bar{A}_{ij} : i \in I_p\}\}$ is constant over all $j \in J_q$.

Here, we slightly abuse the notation not to distinguish $\bar{G}$ and the MILP-graph obtained from $\bar{G}$ by relabelling vertice in $V$, and these two graphs have exactly the same SB scores since the vertices in $W$ are not relabelled.

Note that both $G$ and $\bar{G}$ are MP-tractable, i.e., for any $p \in \{1, 2, \ldots, s\}$ and $q \in \{1, 2, \ldots, t\}$, $(A_{ij})_{i \in I_p, j \in J_q}$ and $(\bar{A}_{ij})_{i \in I_p, j \in J_q}$ are both matrices with identical entries, which combined with the third and the fourth conditions above implies that $A_{ij} = \bar{A}_{ij}$ for all $i \in I_p$ and $j \in J_q$. Therefore, we have $G = \bar{G}$ and hence $\mathrm{SB}(G) = \mathrm{SB}(\bar{G})$.

(b) The result is a directly corollary of (a) by considering $G$ and the MILP-graph obtained from $G$ by relabeling $j_1$ as $j_2$ and relabeling $j_2$ as $j_1$. □

In addition to Theorem A.3, we also need the following two theorem to prove Theorem 4.4.

**Theorem A.4** (Lusin's theorem [16, Theorem 1.14]). *Suppose that $\mu$ is a Borel regular measure on $\mathbb{R}^n$ and that $f : \mathbb{R}^n \to \mathbb{R}^m$ is $\mu$-measurable, i.e., for any open subset $U \subset \mathbb{R}^m$, $f^{-1}(U)$ is $\mu$-measurable. Then for any $\mu$-measurable $X \subset \mathbb{R}^n$ with $\mu(X) < \infty$ and any $\epsilon > 0$, there exists a compact set $E \subset X$ with $\mu(X \backslash E) < \epsilon$, such that $f|_E$ is continuous.*

**Theorem A.5** ([11, Theorem E.1]). *Let $X \subset \mathcal{G}_{m,n}$ be a compact subset that is closed under the action of $S_m \times S_n$. Suppose that $\Phi \in \mathcal{C}(X, \mathbb{R}^n)$ satisfies the followings:*

(a) *For any $\sigma_V \in S_m, \sigma_W \in S_n$, and $G \in X$, it holds that $\Phi((\sigma_V, \sigma_W) * G) = \sigma_W(\Phi(G))$, where $(\sigma_V, \sigma_W) * G$ represents the MILP-graph obtained from $G$ by reordering vertices with permutations $\sigma_V$ and $\sigma_W$.*

(b) *$\Phi(G) = \Phi(\bar{G})$ holds for all $G, \hat{G} \in X$ with $G \overset{W}{\sim} \bar{G}$.*

(c) *Given any $G \in X$ and any $j_1, j_2 \in \{1, 2, \ldots, n\}$, if $C_L^W(j_1) = C_L^W(j_2)$ holds for any $L$ and any hash functions, then $\Phi(G)_{j_1} = \Phi(G)_{j_2}$.*

*Then for any $\epsilon > 0$, there exists $F \in \mathcal{F}_{\text{MP-GNN}}$ such that*

$$\sup_{G \in X} \|\Phi(G) - F(G)\| < \epsilon.$$

Now we can present the proof of Theorem 4.4.

*Proof of Theorem 4.4.* Lemma F.2 and Lemma F.3 in [11] prove that the function that maps LP instances to its optimal objective value/optimal solution with the smallest $\ell_2$-norm is Borel measurable. Thus, $\text{SB} : \mathcal{G}_{m,n} \supset \text{SB}^{-1}(\mathbb{R}^n) \to \mathbb{R}^n$ is also Borel measurable, and is hence $\mathbb{P}$-measurable due to Assumption 4.3. In addition, $\mathcal{G}_{m,n}^{\text{MP}}$ is a Borel subset of $\mathcal{G}_{m,n}$ since the MP-tractability is defined by finitely many operations of comparison and aggregations. By Theorem A.4 and the assumption $\mathbb{P}[G \in \mathcal{G}_{m,n}^{\text{MP}}] = 1$, there exists a compact subset $X_1 \subset \mathcal{G}_{m,n}^{\text{MP}} \cap \text{SB}^{-1}(\mathbb{R}^n)$ such that $\mathbb{P}[\mathcal{G}_{m,n} \backslash X_1] \leq \epsilon$ and $\text{SB}|_{X_1}$ is continuous. For any $\sigma_V \in S_m$ and $\sigma_W \in S_n$, $(\sigma_V, \sigma_W) * X_1$ is also compact and $\text{SB}|_{(\sigma_V, \sigma_W) * X_1}$ is also continuous by the permutation-equivariance of SB. Set

$$X_2 = \bigcup_{\sigma_V \in S_m, \sigma_W \in S_n} (\sigma_V, \sigma_W) * X_1.$$

Then $X_2$ is permutation-invariant and compact with

$$\mathbb{P}[\mathcal{G}_{m,n} \backslash X_2] \leq \mathbb{P}[\mathcal{G}_{m,n} \backslash X_1] \leq \epsilon.$$

In addition, $\text{SB}|_{X_2}$ is continuous by pasting lemma.

The rest of the proof is to apply Theorem A.5 for $X = X_2$ and $\Phi = \text{SB}$, for which we need to verify the four conditions in Theorem C.10. Condition (a) is true since SB is permutation-equivalent by its definition. Conditions (b) and (c) follow directly from Theorem A.3. According to Theorem A.5, there exists some $F \in \mathcal{F}_{\text{2-FGNN}}$ such that

$$\sup_{G \in X_2} \|F(G) - \text{SB}(G)\| \leq \delta.$$

Therefore, one has

$$\mathbb{P}[\|F(G) - \text{SB}(G)\| > \delta] \leq \mathbb{P}[\mathcal{G}_{m,n} \backslash X_2] \leq \epsilon,$$

which completes the proof. □

## B Proof of Theorem 4.5

In this section, we verify that the MILP instances (4.1) and (4.2) prove Theorem 4.5. We will first show that they have different SB scores while cannot be distinguished by any MP-GNNs.

**Different SB scores**    Denote the graph representation of (4.1) and (4.2) as $G$ and $\bar{G}$, respectively. For both (4.1) and (4.2), the same optimal objective value is $4$ and the optimal solution with the smallest $\ell_2$-norm is $(1/2, 1/2, 1/2, 1/2, 1/2, 1/2, 1/2, 1/2)$. To calculate $\mathrm{SB}(G)_j$ or $\mathrm{SB}(\bar{G})_j$, it is necessary create two LPs for each variable $x_j$. In one LP, the upper bound of $x_j$ is set to $\hat{u}_j = \lfloor 1/2 \rfloor = 0$, actually fixing $x_j$ at its lower bound $l_j = 0$. Similarly, the other LP sets $x_j$ to $1$.

For the problem (4.1), even if we fix $x_1 = 1$, the objective value of the LP relaxation can still achieve $4$ by $x = (1, 0, 1, 0, 1, 0, 1, 0)$. A similar observation also holds for fixing $x_1 = 0$. Therefore, the SB score for $x_1$ (also for any $x_j$ in (4.1)) is $0$. In other words,

$$\mathrm{SB}(G) = (0, 0, 0, 0, 0, 0, 0, 0).$$

However, for the problem (4.2), if we fix $x_1 = 1$, then the optimal objective value of the LP relaxation is $9/2$ since

$$\sum_{i=1}^{8} x_i = 1 + (x_2 + x_3) + \frac{1}{2}(x_4 + x_5) + \frac{1}{2}(x_5 + x_6) + \frac{1}{2}(x_6 + x_4) + (x_7 + x_8) \geq 9/2$$

and the above inequality is tight as $x = (1, 1/2, 1/2, 1/2, 1/2, 1/2, 1/2, 1/2)$. If we fix $x_1 = 0$, then $x_2, x_3 \geq 1$ and the optimal objective value of the LP relaxation is also $9/2$ since

$$\sum_{i=1}^{8} x_i \geq 0 + 1 + 1 + \frac{1}{2}(x_4 + x_5) + \frac{1}{2}(x_5 + x_6) + \frac{1}{2}(x_6 + x_4) + (x_7 + x_8) \geq 9/2,$$

and the equality holds when $x = (0, 1, 1, 1/2, 1/2, 1/2, 1/2, 1/2)$. Therefore, the the SB score for $x_1$ (also for any $x_i$ ($1 \leq i \leq 6$) in (4.2)) is $(9/2 - 4) \cdot (9/2 - 4) = 1/4$. If we fix $x_7 = 1$, the optimal objective value of the LP relaxation is still $4$ since $(1/2, 1/2, 1/2, 1/2, 1/2, 1/2, 1, 0)$ is an optimal solution. A similar observation still holds if $x_7$ is fixed to $0$. Thus the SB scores for $x_7$ and $x_8$ are both $0$. Combining these calculations, we obtain that

$$\mathrm{SB}(\bar{G}) = \left(\frac{1}{4}, \frac{1}{4}, \frac{1}{4}, \frac{1}{4}, \frac{1}{4}, \frac{1}{4}, 0, 0\right).$$

**MP-GNNs' output**    Although $G$ and $\bar{G}$ are non-isomorphic with different SB scores, they still have the same output for every MP-GNN. We prove this by induction. Referencing the graph representations in Section 2, we explicitly write down the features:

$$v_i = \bar{v}_i = (1, \geq), \quad w_j = \bar{w}_j = (1, 0, 1, 1), \quad \text{for all } i \in \{1, \cdots, 8\}, \ j \in \{1, \cdots, 8\}.$$

Considering the MP-GNN's initial step where $s_i^0 = p^0(v_i)$ and $t_j^0 = q^0(w_j)$, we can conclude that $s_i^0 = \bar{s}_i^0$ is a constant for all $i$ and $t_j^0 = \bar{t}_j^0$ is a constant for all $j$, regardless of the choice of functions $p^0$ and $q^0$. Thus, the initial layer generates uniform outcomes for nodes in $V$ and $W$ across both graphs, which is the induction base. Suppose that the principle of uniformity applies to $s_i^l, \bar{s}_i^l, t_j^l, \bar{t}_j^l$ for some $0 \leq l \leq L - 1$. Since $s_i^l, \bar{s}_i^l$ are constant across all $i$, we can denote their common value as $s^l$ and hence $s^l = s_i^l = \bar{s}_i^l$ for all $i$. Similarly, we can define $t^l$ with $t^l = t_j^l = \bar{t}_j^l$ for all $j$. Then it holds that

$$s_i^{l+1} = \bar{s}_i^{l+1} = p^l\left(s^l, 2f^l(t^l, 1)\right) \ \text{ and } \ t_j^{l+1} = \bar{t}_j^{l+1} = q^l\left(t^l, 2g^l(s^l, 1)\right),$$

where we used $\{\{A_{ij'} : j' \in W\}\} = \{\{\bar{A}_{ij'} : j' \in W\}\} = \{\{A_{i'j} : i' \in W\}\} = \{\{\bar{A}_{i'j} : i' \in W\}\} = \{\{1, 1, 0, 0, 0, 0, 0, 0\}\}$ for all $i$ and $j$. This proves the uniformity for $l + 1$. Therefore, we obtain the existence of $s^L, t^L$ such that $s_i^L = \bar{s}_i^L = s^L$ and $t_j^L = \bar{t}_j^L = t^L$ for all $i, j$. Finally, the output layer yields:

$$y_j = \bar{y}_j = r\left(8s^L, 8t^L, t^L\right) \quad \text{for all } j \in \{1, \cdots, 8\},$$

which finishes the proof.

## C    Proof of Theorem 4.7

This section presents the proof of Theorem 4.7. The central idea is to establish a separation result in the sense that *two MILPs with distinct SB scores must be distinguished by at least one $F \in \mathcal{F}_{\text{2-FGNN}}$*, and then apply a generalized Stone-Weierstrass theorem in [4].

## C.1  2-FWL test and its separation power

The 2-FWL test [9], as an extension to the classic WL test [53], is a more powerful algorithm for the graph isomorphism problem. By applying the 2-FWL test algorithm (formally stated in Algorithm 2) to two graphs and comparing the outcomes, one can determine the non-isomorphism of the two graphs if the results vary. However, identical 2-FWL outcomes do not confirm isomorphism. Although this test does not solve the graph isomorphism problem entirely, it can serve as a measure of 2-FGNN's separation power, analogous to how the WL test applies to MP-GNN [55].

---

**Algorithm 2** 2-FWL test for MILP-Graphs

---

1: **Input:** A graph instance $G = (V, W, A, F_V, F_W)$ and iteration limit $L > 0$.
2: Initialize with

$$C_0^{VW}(i, j) = \text{HASH}_0^{VW}(v_i, w_j, A_{ij}),$$
$$C_0^{WW}(j_1, j_2) = \text{HASH}_0^{WW}(w_{j_1}, w_{j_2}, \delta_{j_1 j_2}).$$

3: **for** $l = 1, 2, \ldots, L$ **do**
4:    Refine the color

$$C_l^{VW}(i, j) = \text{HASH}_l^{VW}\left(C_{l-1}^{VW}(i, j), \left\{\left\{(C_{l-1}^{WW}(j_1, j), C_{l-1}^{VW}(i, j_1)) : j_1 \in W\right\}\right\}\right),$$
$$C_l^{WW}(j_1, j_2) = \text{HASH}_l^{WW}\left(C_{l-1}^{WW}(j_1, j_2), \left\{\left\{(C_{l-1}^{VW}(i, j_2), C_{l-1}^{VW}(i, j_1)) : i \in V\right\}\right\}\right).$$

5: **end for**
6: **Output:** Final colors $C_L^{VW}(i, j)$ for all $i \in V, j \in W$ and $C_L^{WW}(j_1, j_2)$ for all $j_1, j_2 \in W$.

---

In particular, given the input graph $G$, the 2-FWL test assigns a color for every pair of nodes in the form of $(i, j)$ with $i \in V, j \in W$ or $(j_1, j_2)$ with $j_1, j_2 \in W$. The initial colors are assigned based on the input features and the colors are refined to subcolors at each iteration in the way that two node pairs are of the same subcolor if and only if they have the same color and the same neighbors' color information. Here, the neighborhood of $(i, j)$ involves $\{\{((j_1, j), (i, j_1)) : j_1 \in W\}\}$ and the neighborhood of $(j_1, j_2)$ involves $\{\{((i, j_2), (i, j_1)) : i \in V\}\}$. After sufficient iterations, the final colors are determined. If the final color multisets of two graphs $G$ and $\bar{G}$ are identical, they are deemed indistinguishable by the 2-FWL test, denoted by $G \sim_2 \bar{G}$. One can formally define the separation power of 2-FWL test via two equivalence relations on $\mathcal{G}_{m,n}$ as follows.

**Definition C.1.** *Let $G, \bar{G} \in \mathcal{G}_{m,n}$ and let $C_l^{VW}(i, j), C_l^{WW}(j_1, j_2)$ and $\bar{C}_l^{VW}(i, j), \bar{C}_l^{WW}(j_1, j_2)$ be the colors generated by 2-FWL test for $G$ and $\bar{G}$.*

*(a) We define $G \sim_2 \bar{G}$ if the followings hold for any $L$ and any hash functions:*

$$\{\{C_L^{VW}(i, j) : i \in V, j \in W\}\} = \{\{\bar{C}_L^{VW}(i, j) : i \in V, j \in W\}\}, \qquad \text{(C.1)}$$
$$\{\{C_L^{WW}(j_1, j_2) : j_1, j_2 \in W\}\} = \{\{\bar{C}_L^{WW}(j_1, j_2) : j_1, j_2 \in W\}\}. \qquad \text{(C.2)}$$

*(b) We define $G \overset{W}{\sim}_2 \bar{G}$ if the followings hold for any $L$ and any hash functions:*

$$\{\{C_L^{VW}(i, j) : i \in V\}\} = \{\{\bar{C}_L^{VW}(i, j) : i \in V\}\}, \quad \forall j \in W, \qquad \text{(C.3)}$$
$$\{\{C_L^{WW}(j_1, j) : j_1 \in W\}\} = \{\{\bar{C}_L^{WW}(j_1, j) : j_1 \in W\}\}, \quad \forall j \in W. \qquad \text{(C.4)}$$

It can be seen that (C.3) and (C.4) are stronger than (C.1) and (C.2), since the latter requires that the entire color multiset is the same while the former requires that the color multiset associated with every $j \in W$ is the same. However, we can show that they are equivalent up to a permutation.

**Theorem C.2.** *For any $G, \bar{G} \in \mathcal{G}_{m,n}$, $G \sim_2 \bar{G}$ if and only if there exists a permutation $\sigma_W \in S_n$ such that $G \overset{W}{\sim}_2 \sigma_W * \bar{G}$, where $\sigma_W * \bar{G}$ is the graph obtained by relabeling vertices in $W$ using $\sigma_W$.*

One can understand that both $G \sim_2 \bar{G}$ and $G \overset{W}{\sim}_2 \bar{G}$ mean that $G$ and $\bar{G}$ cannot be distinguished by 2-FWL test, with the difference that $G \sim_2 \bar{G}$ allows a permutation on $W$.

*Proof of Theorem C.2.* It is clear that $G \overset{W}{\sim}_2 \sigma_W * \bar{G}$ implies that $G \sim_2 \bar{G}$. We then prove the reverse direction, i.e., $G \sim_2 \bar{G}$ implies $G \overset{W}{\sim}_2 \sigma_W * \bar{G}$ for some $\sigma_W \in S_n$. It suffices to consider $L$ and hash functions such that there are no collisions in Algorithm 2 and no strict color refinement in the $L$-th iteration when $G$ and $\bar{G}$ are the input, which means that two edges are assigned with the same color in the $L$-th iteration if and only if their colors are the same in the $(L-1)$-th iteration. For any $j_1, j_2, j_1', j_2' \in W$, it holds that

$$C_L^{WW}(j_1, j_2) = C_L^{WW}(j_1', j_2')$$
$$\implies \{\{(C_L^{VW}(i, j_2), C_L^{VW}(i, j_1)) : i \in V\}\} = \{\{(C_L^{VW}(i, j_2'), C_L^{VW}(i, j_1')) : i \in V\}\}$$
$$\implies \{\{C_L^{VW}(i, j_1) : i \in V\}\} = \{\{C_L^{VW}(i, j_1') : i \in V\}\} \text{ and}$$
$$\{\{C_L^{VW}(i, j_2) : i \in V\}\} = \{\{C_L^{VW}(i, j_2') : i \in V\}\}.$$

Similarly, one has that

$$C_L^{WW}(j_1, j_2) = \bar{C}_L^{WW}(j_1', j_2')$$
$$\implies \{\{C_L^{VW}(i, j_1) : i \in V\}\} = \{\{\bar{C}_L^{VW}(i, j_1') : i \in V\}\} \text{ and}$$
$$\{\{C_L^{VW}(i, j_2) : i \in V\}\} = \{\{\bar{C}_L^{VW}(i, j_2') : i \in V\}\},$$

and that

$$\bar{C}_L^{WW}(j_1, j_2) = \bar{C}_L^{WW}(j_1', j_2')$$
$$\implies \{\{\bar{C}_L^{VW}(i, j_1) : i \in V\}\} = \{\{\bar{C}_L^{VW}(i, j_1') : i \in V\}\} \text{ and}$$
$$\{\{\bar{C}_L^{VW}(i, j_2) : i \in V\}\} = \{\{\bar{C}_L^{VW}(i, j_2') : i \in V\}\}.$$

Therefore, for any

$$\mathbf{C} \in \{\{\{C_L^{VW}(i, j) : i \in V\}\} : j \in W\} \cup \{\{\{\bar{C}_L^{VW}(i, j) : i \in V\}\} : j \in W\},$$

it follows from (C.2) that

$$\{\{C_L^{WW}(j_1, j_2) : \{\{C_L^{VW}(i, j_1) : i \in V\}\} = \{\{C_L^{VW}(i, j_2) : i \in V\}\} = \mathbf{C}\}\}$$
$$= \{\{\bar{C}_L^{WW}(j_1, j_2) : \{\{\bar{C}_L^{VW}(i, j_1) : i \in V\}\} = \{\{\bar{C}_L^{VW}(i, j_2) : i \in V\}\} = \mathbf{C}\}\}. \tag{C.5}$$

Particularly, the number of elements in the two multisets in (C.5) should be the same, which implies that

$$\# \{j \in W : \{\{C_L^{VW}(i, j) : i \in V\}\} = \mathbf{C}\} = \# \{j \in W : \{\{\bar{C}_L^{VW}(i, j) : i \in V\}\} = \mathbf{C}\},$$

which then leads to

$$\{\{\{\{C_L^{VW}(i, j) : i \in V\}\} : j \in W\}\} = \{\{\{\{\bar{C}_L^{VW}(i, j) : i \in V\}\} : j \in W\}\}.$$

One can hence apply some permutation on $W$ to obtain (C.3). Next we prove (C.4). For any $j \in W$, we have

$$\{\{C_L^{VW}(i, j) : i \in V\}\} = \{\{\bar{C}_L^{VW}(i, j) : i \in V\}\}$$
$$\implies C_L^{VW}(i_1, j) = \bar{C}_L^{VW}(i_2, j) \quad \text{for some } i_1, i_2 \in V$$
$$\implies \{\{(C_L^{WW}(j_1, j), C_{l-1}^{VW}(i_1, j_1)) : j_1 \in W\}\} = \{\{(\bar{C}_L^{WW}(j_1, j), \bar{C}_{l-1}^{VW}(i_2, j_1)) : j_1 \in W\}\}$$
$$\qquad \text{for some } i_1, i_2 \in V$$
$$\implies \{\{C_L^{WW}(j_1, j) : j_1 \in W\}\} = \{\{\bar{C}_L^{WW}(j_1, j) : j_1 \in W\}\},$$

which completes the proof. $\square$

## C.2 SB scores of MILPs distinguishable by 2-FWL test

The following theorem establishes that the separation power of 2-FWL test is stronger than or equal to that of SB, in the sense that two MILP-graphs, or two vertices in a single graph, that cannot be distinguished by the 2-FWL test must share the same SB score.

**Theorem C.3.** *For any $G, \bar{G} \in \mathcal{G}_{m,n}$, the followings are true:*

(a) If $G \overset{W}{\sim}_2 \bar{G}$, then $\mathrm{SB}(G) = \mathrm{SB}(\bar{G})$.

(b) If $G \sim_2 \bar{G}$, then there exists some permutation $\sigma_W \in S_n$ such that $\mathrm{SB}(G) = \sigma_W(\mathrm{SB}(\bar{G}))$.

(c) If $\{\{C_L^{WW}(j, j_1) : j \in W\}\} = \{\{C_L^{WW}(j, j_2) : j \in W\}\}$ holds for any $L$ and any hash functions, then $\mathrm{SB}(G)_{j_1} = \mathrm{SB}(G)_{j_2}$.

We briefly describe the intuition behind the proof here. The color updating rule of 2-FWL test is based on monitoring triangles while that of the classic WL test is based on tracking edges. More specifically, in 2-FWL test colors are defined on node pairs and neighbors share the same triangle, while in WL test colors are equipped with nodes with neighbors being connected by edges. When computing the $j$-th entry of $\mathrm{SB}(G)$, we change the upper/lower bound of $x_j$ and solve two LP problems. We can regard $j \in W$ as a special node and if we fixed it in 2-FWL test, a triangle containing $j \in W$ will be determined by the other two nodes, one in $V$ and one in $W$, and their edge. This "reduces" to the setting of WL test. It is proved in [11] that the separation power of WL test is stronger than or equal to the properties of LPs. This is to say that even when fixing a special node, the 2-FWL test still has enough separation power to distinguish different LP properties and hence 2-FWL test could separate different SB scores. We present the detailed proof of Theorem C.3 in the rest of this subsection.

**Theorem C.4.** *For any $G, \bar{G} \in \mathcal{G}_{m,n}$, if $G \overset{W}{\sim}_2 \bar{G}$, then for any $j \in \{1, 2, \ldots, n\}$, $\hat{l}_j \in \{-\infty\} \cup \mathbb{R}$, and $\hat{u}_j \in \mathbb{R} \cup \{+\infty\}$, the two LP problems $\mathrm{LP}(G, j, \hat{l}_j, \hat{u}_j)$ and $\mathrm{LP}(\bar{G}, j, \hat{l}_j, \hat{u}_j)$ have the same optimal objective value.*

**Theorem C.5** ([11]). *Consider two LP problems with $n$ variables and $m$ constraints*

$$\min_{x \in \mathbb{R}^n} \quad c^\top x, \quad \text{s.t.} \quad Ax \circ b, \; l \leq x \leq u, \tag{C.6}$$

*and*

$$\min_{x \in \mathbb{R}^n} \quad \bar{c}^\top x, \quad \text{s.t.} \quad \bar{A}x \bar{\circ} \bar{b}, \; \bar{l} \leq x \leq \bar{u}. \tag{C.7}$$

*Suppose that there exist $\mathcal{I} = \{I_1, I_2, \ldots, I_s\}$ and $\mathcal{J} = \{J_1, J_2, \ldots, J_t\}$ that are partitions of $V = \{1, 2, \ldots, m\}$ and $W = \{1, 2, \ldots, n\}$ respectively, such that the followings hold:*

(a) *For any $p \in \{1, 2, \ldots, s\}$, $(b_i, \circ_i) = (\bar{b}_i, \bar{\circ}_i)$ is constant over all $i \in I_p$;*

(b) *For any $q \in \{1, 2, \ldots, t\}$, $(c_j, l_j, u_j) = (\bar{c}_j, \bar{l}_j, \bar{u}_j)$ is constant over all $j \in J_q$;*

(c) *For any $p \in \{1, 2, \ldots, s\}$ and $q \in \{1, 2, \ldots, t\}$, $\sum_{j \in J_q} A_{ij} = \sum_{j \in J_q} \bar{A}_{ij}$ is constant over all $i \in I_p$.*

(d) *For any $p \in \{1, 2, \ldots, s\}$ and $q \in \{1, 2, \ldots, t\}$, $\sum_{i \in I_p} A_{ij} = \sum_{i \in I_p} \bar{A}_{ij}$ is constant over all $j \in J_q$.*

*Then the two problems (C.6) and (C.7) have the same feasibility, the same optimal objective value, and the same optimal solution with the smallest $\ell_2$-norm (if feasible and bounded).*

*Proof of Theorem C.4.* Choose $L$ and hash functions such that there are no collisions in Algorithm 2 and no strict color refinement in the $L$-th iteration when $G$ and $\bar{G}$ are the input. Fix any $j \in W$ and construct the partitions $\mathcal{I} = \{I_1, I_2, \ldots, I_s\}$ and $\mathcal{J} = \{J_1, J_2, \ldots, J_t\}$ as follows:

- $i_1, i_2 \in I_p$ for some $p \in \{1, 2, \ldots, s\}$ if and only if $C_L^{VW}(i_1, j) = C_L^{VW}(i_2, j)$.

- $j_1, j_2 \in J_q$ for some $q \in \{1, 2, \ldots, t\}$ if and only if $C_L^{WW}(j_1, j) = C_L^{WW}(j_2, j)$.

Without loss of generality, we can assume that $j \in J_1$. One observation is that $J_1 = \{j\}$. This is because $j_1 \in J_1$ implies that $C_L^{WW}(j_1, j) = C_L^{WW}(j, j)$, which then leads to $C_0^{WW}(j_1, j) = C_0^{WW}(j, j)$ and $\delta_{j_1 j} = \delta_{jj} = 1$ since there is no collisions. We thus have $j_1 = j$.

Note that we have (C.3) and (C.4) from the assumption $G \overset{W}{\sim}_2 \bar{G}$. So after permuting $\bar{G}$ on $V$ and $W \backslash \{j\}$, one can obtain $C_L^{VW}(i, j) = \bar{C}_L^{VW}(i, j)$ for all $i \in V$ and $C_L^{WW}(j_1, j) = \bar{C}_L^{WW}(j_1, j)$ for all $j_1 \in W$. Another observation is that such permutation does not change the optimal objective value of $\mathrm{LP}(\bar{G}, j, \hat{l}_j, \hat{u}_j)$ as $j$ is fixed.

Next, we verify the four conditions in Theorem C.5 for two LP problems $\text{LP}(G, j, \hat{l}_j, \hat{u}_j)$ and $\text{LP}(\bar{G}, j, \hat{l}_j, \hat{u}_j)$ with respect to the partitions $\mathcal{I} = \{I_1, I_2, \ldots, I_s\}$ and $\mathcal{J} = \{J_1, J_2, \ldots, J_t\}$.

**Verification of Condition (a) in Theorem C.5**  Since there is no collision in the 2-FWL test Algorithm 2, $C_L^{VW}(i, j) = \bar{C}_L^{VW}(i, j)$ implies that $C_0^{VW}(i, j) = \bar{C}_0^{VW}(i, j)$ and hence that $v_i = \bar{v}_i$, which is also constant over all $i \in I_p$ since $C_L^{VW}(i, j)$ is contant over all $i \in I_p$ by definition.

**Verification of Condition (b) in Theorem C.5**  It follows from $C_L^{WW}(j_1, j) = \bar{C}_L^{WW}(j_1, j)$ that $C_0^{WW}(j_1, j) = \bar{C}_0^{WW}(j_1, j)$ and hence that $w_{j_1} = \bar{w}_{j_1}$, which is also constant over all $j_1 \in I_q$ since $C_L^{WW}(j_1, j)$ is contant over all $j_1 \in I_q$ by definition.

**Verification of Condition (c) in Theorem C.5**  Consider any $p \in \{1, 2, \ldots, s\}$ and any $i \in I_p$. It follows from $C_L^{VW}(i, j) = \bar{C}_L^{VW}(i, j)$ that

$$\left\{\left\{(C_{L-1}^{WW}(j_1, j), C_{L-1}^{VW}(i, j_1)) : j_1 \in W\right\}\right\} = \left\{\left\{(\bar{C}_{L-1}^{WW}(j_1, j), \bar{C}_{L-1}^{VW}(i, j_1)) : j_1 \in W\right\}\right\},$$

and hence that

$$\left\{\left\{(C_L^{WW}(j_1, j), A_{ij_1}) : j_1 \in W\right\}\right\} = \left\{\left\{(\bar{C}_L^{WW}(j_1, j), \bar{A}_{ij_1}) : j_1 \in W\right\}\right\},$$

where we used the fact that there is no strict color refinement in the $L$-th iteration and there is no collision in Algorithm 2. We can thus conclude for any $q \in \{1, 2, \ldots, t\}$ that

$$\{\{A_{ij_1} : j_1 \in J_q\}\} = \{\{\bar{A}_{ij_1} : j_1 \in J_q\}\},$$

which implies that $\sum_{j_1 \in J_q} A_{ij_1} = \sum_{j_1 \in J_q} \bar{A}_{ij_1}$ that is constant over $i \in I_p$ since $C_L^{VW}(i, j) = \bar{C}_L^{VW}(i, j)$ is constant over $i \in I_p$.

**Verification of Condition (d) in Theorem C.5**  Consider any $q \in \{1, 2, \ldots, t\}$ and any $j_1 \in J_q$. It follows from $C_L^{WW}(j_1, j) = \bar{C}_L^{WW}(j_1, j)$ that

$$\left\{\left\{(C_{L-1}^{VW}(i, j), C_{L-1}^{VW}(i, j_1)) : i \in V\right\}\right\} = \left\{\left\{(\bar{C}_{L-1}^{VW}(i, j), \bar{C}_{L-1}^{VW}(i, j_1)) : i \in V\right\}\right\},$$

and hence that

$$\left\{\left\{(C_L^{VW}(i, j), A_{ij_1}) : i \in V\right\}\right\} = \left\{\left\{(\bar{C}_L^{VW}(i, j), \bar{A}_{ij_1}) : i \in V\right\}\right\},$$

where we used the fact that there is no strict color refinement at the $L$-th iteration and there is no collision in Algorithm 2. We can thus conclude for any $p \in \{1, 2, \ldots, s\}$ that

$$\{\{A_{ij_1} : i \in I_p\}\} = \{\{\bar{A}_{ij_1} : i \in I_p\}\},$$

which implies that $\sum_{i \in I_p} A_{ij_1} = \sum_{i \in I_p} \bar{A}_{ij_1}$ that is constant over $j_1 \in J_q$ since $C_L^{WW}(j_1, j) = \bar{C}_L^{WW}(j_1, j)$ is constant over $j_1 \in J_q$.

Combining all discussion above and noticing that $J_1 = \{j\}$, one can apply Theorem C.5 and conclude that the two LP problems $\text{LP}(G, j, \hat{l}_j, \hat{u}_j)$ and $\text{LP}(\bar{G}, j, \hat{l}_j, \hat{u}_j)$ have the same optimal objective value, which completes the proof. $\qquad\square$

**Corollary C.6.** *For any $G, \bar{G} \in \mathcal{G}_{m,n}$, if $G \overset{W}{\sim}_2 \bar{G}$, then the LP relaxations of $G$ and $\bar{G}$ have the same optimal objective value and the same optimal solution with the smallest $\ell_2$-norm (if feasible and bounded).*

*Proof.* If no collision, it follows from (C.4) that $C_L^{WW}(j, j) = \bar{C}_L^{WW}(j, j)$ which implies $l_j = \bar{l}_j$ and $u_j = \bar{u}_j$ for any $j \in W$. Then we can apply Theorem C.4 to conclude that two LP problems $\text{LP}(G, j, l_j, u_j)$ and $\text{LP}(\bar{G}, j, \bar{l}_j, \bar{u}_j)$ that are LP relaxations of $G$ and $\bar{G}$ have the same optimal objective value.

In the case that the LP relaxations of $G$ and $\bar{G}$ are both feasible and bounded, we use $x$ and $\bar{x}$ to denote their optimal solutions with the smallest $\ell_2$-norm. For any $j \in W$, $x$ and $\bar{x}$ are also the optimal solutions with the smallest $\ell_2$-norm for $\text{LP}(G, j, l_j, u_j)$ and $\text{LP}(\bar{G}, j, \bar{l}_j, \bar{u}_j)$ respectively. By Theorem C.5 and the same arguments as in the proof of Theorem C.4, we have the $x_j = \bar{x}_j$. Note that we cannot infer $x = \bar{x}$ by considering a single $j \in W$ because we apply permutation on $V$ and $W \setminus \{j\}$ in the proof of Theorem C.4. But we have $x_j = \bar{x}_j$ for any $j \in W$ which leads to $x = \bar{x}$. $\quad\square$

*Proof of Theorem C.3.* (a) By Corollary C.6 and Theorem C.4.

(b) By Theorem C.2 and (a).

(c) Apply (a) on $G$ and the graph obtained from $G$ by switching $j_1$ and $j_2$. $\qquad\square$

## C.3 Equivalence between the separation powers of the 2-FWL test and 2-FGNNs

The section establishes the equivalence between the separation powers of the 2-FWL test and 2-FGNNs.

**Theorem C.7.** *For any $G, \bar{G} \in \mathcal{G}_{m,n}$, the followings are true:*

(a) $G \overset{W}{\sim}_2 \bar{G}$ *if and only if $F(G) = F(\bar{G})$ for all $F \in \mathcal{F}_{\text{2-FGNN}}$.*

(b) $\{\{C_L^{WW}(j, j_1) : j \in W\}\} = \{\{C_L^{WW}(j, j_2) : j \in W\}\}$ *holds for any $L$ and any hash functions if and only if $F(G)_{j_1} = F(G)_{j_2}$, $\forall F \in \mathcal{F}_{\text{2-FGNN}}$.*

(c) $G \sim_2 \bar{G}$ *if and only if $f(G) = f(\bar{G})$ for all scalar function $f$ with $f\mathbf{1} \in \mathcal{F}_{\text{2-FGNN}}$.*

The intuition behind Theorem C.7 is the color updating rule in 2-FWL test is of the same format as the feature updating rule in 2-FGNN, and that the local update mappings $p^l, q^l, f^l, g^l, r$ can be chosen as injective on current features. Results of similar spirit also exist in previous literature; see e.g., [4, 11, 22, 55]. We present the detailed proof of Theorem C.7 in the rest of this subsection.

**Lemma C.8.** *For any $G, \bar{G} \in \mathcal{G}_{m,n}$, if $G \overset{W}{\sim}_2 \bar{G}$, then $F(G) = F(\bar{G})$ for all $F \in \mathcal{F}_{\text{2-FGNN}}$.*

*Proof.* Consider any $F \in \mathcal{F}_{\text{2-FGNN}}$ with $L$ layers and let $s_{ij}^l, t_{j_1 j_2}^l$ and $\bar{s}_{ij}^l, \bar{t}_{j_1 j_2}^l$ be the features in the $l$-th layer of $F$. Choose $L$ and hash functions such that there are no collisions in Algorithm 2 when $G$ and $\bar{G}$ are the input. We will prove the followings by induction for $0 \le l \le L$:

(a) $C_l^{VW}(i, j) = C_l^{VW}(i', j')$ implies $s_{ij}^l = s_{i'j'}^l$.

(b) $C_l^{VW}(i, j) = \bar{C}_l^{VW}(i', j')$ implies $s_{ij}^l = \bar{s}_{i'j'}^l$.

(c) $\bar{C}_l^{VW}(i, j) = \bar{C}_l^{VW}(i', j')$ implies $\bar{s}_{ij}^l = \bar{s}_{i'j'}^l$.

(d) $C_l^{WW}(j_1, j_2) = C_l^{WW}(j_1', j_2')$ implies $t_{j_1 j_2}^l = t_{j_1' j_2'}^l$.

(e) $C_l^{WW}(j_1, j_2) = \bar{C}_l^{WW}(j_1', j_2')$ implies $t_{j_1 j_2}^l = \bar{t}_{j_1' j_2'}^l$.

(f) $\bar{C}_l^{WW}(j_1, j_2) = \bar{C}_l^{WW}(j_1', j_2')$ implies $\bar{t}_{j_1 j_2}^l = \bar{t}_{j_1' j_2'}^l$.

As the induction base, the claims (a)-(f) are true for $l = 0$ since $\text{HASH}_0^{VW}$ and $\text{HASH}_0^{WW}$ do not have collisions. Now we assume that the claims (a)-(f) are all true for $l - 1$ where $l \in \{1, 2, \ldots, L\}$ and prove them for $l$. In fact, one can prove the claim (a) for $l$ as follow:

$$C_l^{VW}(i, j) = C_l^{VW}(i', j')$$
$$\Longrightarrow C_{l-1}^{VW}(i, j) = C_{l-1}^{VW}(i', j') \quad \text{and}$$
$$\{\{(C_{l-1}^{WW}(j_1, j), C_{l-1}^{VW}(i, j_1)) : j_1 \in W\}\} = \{\{(C_{l-1}^{WW}(j_1, j'), C_{l-1}^{VW}(i', j_1)) : j_1 \in W\}\}$$
$$\Longrightarrow s_{ij}^{l-1} = s_{i'j'}^{l-1} \quad \text{and} \quad \{\{(t_{j_1 j}^{l-1}, s_{ij_1}^{l-1}) : j_1 \in W\}\} = \{\{(t_{j_1 j'}^{l-1}, s_{i'j_1}^{l-1}) : j_1 \in W\}\}$$
$$\Longrightarrow s_{ij}^l = s_{i'j'}^l.$$

The proof of claims (b)-(f) for $l$ is very similar and hence omitted.

Using the claims (a)-(f) for $L$, we can conclude that

$$G \overset{W}{\sim}_2 \bar{G}$$
$$\Longrightarrow \{\{C_L^{VW}(i, j) : i \in V\}\} = \{\{\bar{C}_L^{VW}(i, j) : i \in V\}\}, \ \forall j \in W, \text{ and}$$

$$\{\{C_L^{WW}(j_1,j):j_1 \in W\}\} = \{\{\bar{C}_L^{WW}(j_1,j):j_1 \in W\}\}, \; \forall\, j \in W$$
$$\Longrightarrow \{\{s_{ij}^L : i \in V\}\} = \{\{\bar{s}_{ij}^L : i \in V\}\}, \; \forall\, j \in W, \text{ and}$$
$$\{\{t_{j_1j}^L : j_1 \in W\}\} = \{\{\bar{t}_{j_1j}^L : j_1 \in W\}\}, \; \forall\, j \in W$$
$$\Longrightarrow r\left(\sum_{i \in V} s_{ij}^L, \sum_{j_1 \in W} t_{j_1j}^L\right) = r\left(\sum_{i \in V} \bar{s}_{ij}^L, \sum_{j_1 \in W} \bar{t}_{j_1j}^L\right), \; \forall\, j \in W$$
$$\Longrightarrow F(G) = F(\bar{G}),$$

which completes the proof. $\qquad\square$

**Lemma C.9.** *For any $G, \bar{G} \in \mathcal{G}_{m,n}$, if $F(G) = F(\bar{G})$ for all $F \in \mathcal{F}_{\text{2-FGNN}}$, then $G \overset{W}{\sim}_2 \bar{G}$.*

*Proof.* We claim that for any $L$ there exists 2-FGNN layers for $l = 0,1,2,\ldots,L$, such that the followings hold true for any $0 \le l \le L$ and any hash functions:

(a) $s_{ij}^l = s_{i'j'}^l$ implies $C_l^{VW}(i,j) = C_l^{VW}(i',j')$.

(b) $s_{ij}^l = \bar{s}_{i'j'}^l$ implies $C_l^{VW}(i,j) = \bar{C}_l^{VW}(i',j')$.

(c) $\bar{s}_{ij}^l = \bar{s}_{i'j'}^l$ implies $\bar{C}_l^{VW}(i,j) = \bar{C}_l^{VW}(i',j')$.

(d) $t_{j_1j_2}^l = t_{j_1'j_2'}^l$ implies $C_l^{WW}(j_1,j_2) = C_l^{WW}(j_1',j_2')$.

(e) $t_{j_1j_2}^l = \bar{t}_{j_1'j_2'}^l$ implies $C_l^{WW}(j_1,j_2) = \bar{C}_l^{WW}(j_1',j_2')$.

(f) $\bar{t}_{j_1j_2}^l = \bar{t}_{j_1'j_2'}^l$ implies $\bar{C}_l^{WW}(j_1,j_2) = \bar{C}_l^{WW}(j_1',j_2')$.

Such layers can be constructed inductively. First, for $l = 0$, we can simply choose $p^0$ that is injective on $\{(v_i, w_j, A_{ij}) : i \in V, j \in W\} \cup \{(\bar{v}_i, \bar{w}_j, \bar{A}_{ij}) : i \in V, j \in W\}$ and $q^0$ that is injective on $\{(w_{j_1}, w_{j_2}, \delta_{j_1j_2}) : j_1, j_2 \in W\} \cup \{(\bar{w}_{j_1}, \bar{w}_{j_2}, \delta_{j_1j_2}) : j_1, j_2 \in W\}$.

Assume that the conditions (a)-(f) are true for $l-1$ where $1 \le l \le L$, we aim to construct the $l$-th layer such that (a)-(f) are also true for $l$. Let $\alpha_1, \alpha_2, \ldots, \alpha_u$ collect all different elements in $\{s_{ij}^{l-1} : i \in V, j \in W\} \cup \{\bar{s}_{ij}^{l-1} : i \in V, j \in W\}$ and let $\beta_1, \beta_2, \ldots, \beta_{u'}$ collect all different elements in $\{t_{j_1j_2}^{l-1} : j_1, j_2 \in W\} \cup \{\bar{t}_{j_1j_2}^{l-1} : j_1, j_2 \in W\}$. Choose some continuous $f^l$ such that $f^l(\beta_{k'}, \alpha_k) = e_{k'}^{u'} \otimes e_k^u \in \mathbb{R}^{u' \times u}$, where $e_{k'}^{u'}$ is a vector in $\mathbb{R}^{u'}$ with the $k'$-th entry being 1 and other entries being 0, and $e_k^u$ is a vector in $\mathbb{R}^u$ with the $k$-th entry being 1 and other entries being 0. Choose some continuous $p^l$ that is injective on the set $\left\{s_{ij}^{l-1}, \sum_{j_1 \in W} f^l(t_{j_1j}^{l-1}, s_{ij_1}^{l-1}) : i \in V, j \in W\right\} \cup \left\{\bar{s}_{ij}^{l-1}, \sum_{j_1 \in W} f^l(\bar{t}_{j_1j}^{l-1}, \bar{s}_{ij_1}^{l-1}) : i \in V, j \in W\right\}$. By the injectivity of $p^l$ and the linear independence of $\{e_{k'}^{u'} \otimes e_k^u : 1 \le k \le u, 1 \le k' \le u'\}$, we have that

$$s_{ij}^l = s_{i'j'}^l$$
$$\Longrightarrow s_{ij}^{l-1} = s_{i'j'}^{l-1} \quad \text{and} \quad \sum_{j_1 \in W} f^l(t_{j_1j}^{l-1}, s_{ij_1}^{l-1}) = \sum_{j_1 \in W} f^l(t_{j_1j'}^{l-1}, s_{i'j_1}^{l-1})$$
$$\Longrightarrow s_{ij}^{l-1} = s_{i'j'}^{l-1} \quad \text{and for any } 1 \le k \le u, \; 1 \le k' \le u'$$
$$\#\{j_1 \in W : t_{j_1j}^{l-1} = \beta_{k'}, s_{ij_1}^{l-1} = \alpha_k\} = \#\{j_1 \in W : t_{j_1j'}^{l-1} = \beta_{k'}, s_{i'j_1}^{l-1} = \alpha_k\}$$
$$\Longrightarrow s_{ij}^{l-1} = s_{i'j'}^{l-1} \quad \text{and} \quad \{\{(t_{j_1j}^{l-1}, s_{ij_1}^{l-1}) : j_1 \in W\}\} = \{\{(t_{j_1j'}^{l-1}, s_{i'j_1}^{l-1}) : j_1 \in W\}\}$$
$$\Longrightarrow C_{l-1}^{VW}(i,j) = C_{l-1}^{VW}(i',j') \quad \text{and}$$
$$\left\{\{(C_{l-1}^{WW}(j_1,j), C_{l-1}^{VW}(i,j_1)) : j_1 \in W\}\right\} = \left\{\{(C_{l-1}^{WW}(j_1,j'), C_{l-1}^{VW}(i',j_1)) : j_1 \in W\}\right\}$$
$$\Longrightarrow C_l^{VW}(i,j) = C_l^{VW}(i',j'),$$

which is to say that the condition (a) is satisfied. One can also verify that the conditions (b) and (c) by using the same argument. Similarly, we can also construct $g^l$ and $q^l$ such that the conditions (d)-(f) are satisfied.

Suppose that $G \overset{W}{\sim}_2 \bar{G}$ is not true. Then there exists $L$ and hash functions such that

$$\{\{C_L^{VW}(i,j) : i \in V\}\} \neq \{\{\bar{C}_L^{VW}(i,j) : i \in V\}\},$$

or

$$\{\{C_L^{WW}(j_1,j) : j_1 \in W\}\} \neq \{\{\bar{C}_L^{WW}(j_1,j) : j_1 \in W\}\},$$

holds for some $j \in W$. We have shown above that the conditions (a)-(f) are true for $L$ and some carefully constructed 2-FGNN layers. Then it holds for some $j \in W$ that

$$\{\{s_{ij}^L : i \in V\}\} \neq \{\{\bar{s}_{ij}^L : i \in V\}\}, \tag{C.8}$$

or

$$\{\{t_{j_1 j}^L : j_1 \in W\}\} \neq \{\{\bar{t}_{j_1 j}^L : j_1 \in W\}\}. \tag{C.9}$$

In the rest of the proof we work with (C.8), and the argument can be easily modified in the case that (C.9) is true. It follows from (C.8) that there exists some continuous function $\varphi$ such that

$$\sum_{i \in V} \varphi(s_{ij}^L) \neq \sum_{i \in V} \varphi(\bar{s}_{ij}^L).$$

Then let us construct the $(L+1)$-th layer yielding

$$s_{ij}^{L+1} = \varphi(s_{ij}^L) \quad \text{and} \quad \bar{s}_{ij}^{L+1} = \varphi(\bar{s}_{ij}^L),$$

and the output layer with

$$r\left(\sum_{i \in V} s_{ij}^{L+1}, \sum_{j_1 \in W} t_{j_1 j}^{L+1}\right) = \sum_{i \in V} \varphi(s_{ij}^L) \neq \sum_{i \in V} \varphi(\bar{s}_{ij}^L) = r\left(\sum_{i \in V} \bar{s}_{ij}^{L+1}, \sum_{j_1 \in W} \bar{t}_{j_1 j}^{L+1}\right).$$

This is to say $F(G)_j \neq F(\bar{G})_j$ for some $F \in \mathcal{F}_{\text{2-FGNN}}$, which contradicts the assumtion that $F$ has the same output on $G$ and $\bar{G}$. Thus we can conclude that $G \overset{W}{\sim}_2 \bar{G}$. $\square$

*Proof of Theorem C.7 (a).* By Lemma C.8 and Lemma C.9. $\square$

*Proof of Theorem C.7 (b).* Apply Theorem C.7 on $G$ and the graph obtained from $G$ by switching $j_1$ and $j_2$. $\square$

*Proof of Theorem C.7 (c).* Suppose that $G \sim_2 \bar{G}$. By Theorem C.2, there exists some permutation $\sigma_W \in S_n$ such that $G \overset{W}{\sim}_2 \sigma_W * \bar{G}$. For any scalar function $f$ with $f\mathbf{1} \in \mathcal{F}_{\text{2-FGNN}}$, by Theorem C.7, it holds that $(f\mathbf{1})(G) = (f\mathbf{1})(\sigma_W * \bar{G}) = (f\mathbf{1})(\bar{G})$, where we used the fact that $f\mathbf{1}$ is permutation-equivariant. We can thus conclude that $f(G) = f(\bar{G})$.

Now suppose that $G \sim_2 \bar{G}$ is not true. Then there exist some $L$ and hash functions such that

$$\{\{C_L^{VW}(i,j) : i \in V, j \in W\}\} \neq \{\{\bar{C}_L^{VW}(i,j) : i \in V, j \in W\}\},$$

or

$$\{\{C_L^{WW}(j_1,j_2) : j_1, j_2 \in W\}\} \neq \{\{\bar{C}_L^{WW}(j_1,j_2) : j_1, j_2 \in W\}\}.$$

By the proof of Lemma C.9, one can construct the $l$-th 2-FGNN layers inductively for $0 \leq l \leq L$, such that the condition (a)-(f) in the proof of Lemma C.9 are true. Then we have

$$\{\{s_{ij}^L : i \in V, j \in W\}\} \neq \{\{\bar{s}_{ij}^L : i \in V, j \in W\}\}, \tag{C.10}$$

or

$$\{\{t_{j_1 j_2}^L : j_1, j_2 \in W\}\} \neq \{\{\bar{t}_{j_1 j_2}^L : j_1, j_2 \in W\}\}. \tag{C.11}$$

We first assume that (C.10) is true. Then there exists a continuous function $\varphi$ with

$$\sum_{i \in V, j \in W} \varphi(s_{ij}^L) \neq \sum_{i \in V, j \in W} \varphi(\bar{s}_{ij}^L).$$

Let us construct the $(L+1)$-th layer such that

$$s_{ij}^{L+1} = p^{L+1}\left(s_{ij}^L, \sum_{j_1 \in W} f^{L+1}(t_{j_1j}^L, s_{ij_1}^L)\right) = \sum_{j_1 \in W} \varphi(s_{ij_1}^L),$$

$$\bar{s}_{ij}^{L+1} = p^{L+1}\left(\bar{s}_{ij}^L, \sum_{j_1 \in W} f^{L+1}(\bar{t}_{j_1j}^L, \bar{s}_{ij_1}^L)\right) = \sum_{j_1 \in W} \varphi(\bar{s}_{ij_1}^L),$$

and the output layer with

$$r\left(\sum_{i \in V} s_{ij}^{L+1}, \sum_{j_1 \in W} t_{j_1j}^{L+1}\right) = \sum_{i \in V}\sum_{j_1 \in W} \varphi(s_{ij_1}^L) \neq \sum_{i \in V}\sum_{j_1 \in W} \varphi(\bar{s}_{ij_1}^L) = r\left(\sum_{i \in V} \bar{s}_{ij}^{L+1}, \sum_{j_1 \in W} \bar{t}_{j_1j}^{L+1}\right),$$

which is independent of $j \in W$. This constructs $F \in \mathcal{F}_{\text{2-FGNN}}$ of the form $F = f\mathbf{1}$ with $f(G) \neq f(\bar{G})$.

Next, we consider the case that (C.11) is true. Then

$$\{\{\{\{t_{j_1j_2}^L : j_1 \in W\}\} : j_2 \in W\}\} \neq \{\{\{\{\bar{t}_{j_1j_2}^L : j_1 \in W\}\} : j_2 \in W\}\}, \tag{C.12}$$

and hence there exists some continuous $\psi$ such that

$$\left\{\left\{\sum_{j_1 \in W} \psi(t_{j_1j_2}^L) : j_2 \in W\right\}\right\} \neq \left\{\left\{\sum_{j_1 \in W} \psi(\bar{t}_{j_1j_2}^L) : j_2 \in W\right\}\right\}.$$

Let us construct the $(L+1)$-th layer such that

$$s_{ij}^{L+1} = p^{L+1}\left(s_{ij}^L, \sum_{j_1 \in W} f^{L+1}(t_{j_1j}^L, s_{ij_1}^L)\right) = \sum_{j_1 \in W} \psi(t_{j_1j}^L),$$

$$\bar{s}_{ij}^{L+1} = p^{L+1}\left(\bar{s}_{ij}^L, \sum_{j_1 \in W} f^{L+1}(\bar{t}_{j_1j}^L, \bar{s}_{ij_1}^L)\right) = \sum_{j_1 \in W} \psi(\bar{t}_{j_1j}^L),$$

and we have from (C.12) that

$$\{\{s_{ij}^{L+1} : i \in V, j \in W\}\} \neq \{\{\bar{s}_{ij}^{L+1} : i \in V, j \in W\}\}.$$

We can therefore repeat the argument for (C.10) and show the existence of $f$ with $f\mathbf{1} \in \mathcal{F}_{\text{2-FGNN}}$ and $f(G) \neq f(\bar{G})$. The proof is hence completed. $\qquad\square$

## C.4 Proof of Theorem 4.7

We finalize the proof of Theorem 4.7 in this subsection. Combining Theorem C.3 and Theorem C.7, one can conclude that the separation power of $\mathcal{F}_{\text{2-FGNN}}$ is stronger than or equal to that of SB scores. Hence, we can apply the Stone-Weierstrass-type theorem to prove Theorem 4.7

**Theorem C.10** (Generalized Stone-Weierstrass theorem [4]). *Let $X$ be a compact topology space and let $\mathbf{G}$ be a finite group that acts continuously on $X$ and $\mathbb{R}^n$. Define the collection of all equivariant continuous functions from $X$ to $\mathbb{R}^n$ as follows:*

$$\mathcal{C}_E(X, \mathbb{R}^n) = \{F \in \mathcal{C}(X, \mathbb{R}^n) : F(g * x) = g * F(x), \ \forall \, x \in X, g \in \mathbf{G}\}.$$

*Consider any $\mathcal{F} \subset \mathcal{C}_E(X, \mathbb{R}^n)$ and any $\Phi \in \mathcal{C}_E(X, \mathbb{R}^n)$. Suppose the following conditions hold:*

(a) *$\mathcal{F}$ is a subalgebra of $\mathcal{C}(X, \mathbb{R}^n)$ and $\mathbf{1} \in \mathcal{F}$, where $\mathbf{1}$ is the constant function whose ouput is always $(1, 1, \ldots, 1) \in \mathbb{R}^n$.*

(b) *For any $x, x' \in X$, if $f(x) = f(x')$ holds for any $f \in \mathcal{C}(X, \mathbb{R})$ with $f\mathbf{1} \in \mathcal{F}$, then for any $F \in \mathcal{F}$, there exists $g \in \mathbf{G}$ such that $F(x) = g * F(x')$.*

*(c) For any $x, x' \in X$, if $F(x) = F(x')$ holds for any $F \in \mathcal{F}$, then $\Phi(x) = \Phi(x')$.*

*(d) For any $x \in X$, it holds that $\Phi(x)_{j_1} = \Phi(x)_{j_2}$, $\forall (j_1, j_2) \in J(x)$, where*

$$J(x) = \left\{ (j_1, j_2) \in \{1, 2, \ldots, n\}^2 : F(x)_{j_1} = F(x)_{j_2}, \ \forall F \in \mathcal{F} \right\}.$$

*Then for any $\epsilon > 0$, there exists $F \in \mathcal{F}$ such that*

$$\sup_{x \in X} \|F(x) - \Phi(x)\| \le \epsilon.$$

*Proof of Theorem 4.7.* Lemma F.2 and Lemma F.3 in [11] prove that the function that maps LP instances to its optimal objective value/optimal solution with the smallest $\ell_2$-norm is Borel measurable. Thus, SB $: \mathcal{G}_{m,n} \supset \mathrm{SB}^{-1}(\mathbb{R}^n) \to \mathbb{R}^n$ is also Borel measurable, and is hence $\mathbb{P}$-measurable due to Assumption 4.3. By Theorem A.4 and Assumption 4.3, there exists a compact subset $X_1 \subset \mathrm{SB}^{-1}(\mathbb{R}^n)$ such that $\mathbb{P}[\mathcal{G}_{m,n} \backslash X_1] \le \epsilon$ and $\mathrm{SB}|_{X_1}$ is continuous. For any $\sigma_V \in S_m$ and $\sigma_W \in S_n$, $(\sigma_V, \sigma_W) * X_1$ is also compact and $\mathrm{SB}|_{(\sigma_V, \sigma_W) * X_1}$ is also continuous by the permutation-equivariance of SB. Set

$$X_2 = \bigcup_{\sigma_V \in S_m, \sigma_W \in S_n} (\sigma_V, \sigma_W) * X_1.$$

Then $X_2$ is permutation-invariant and compact with

$$\mathbb{P}[\mathcal{G}_{m,n} \backslash X_2] \le \mathbb{P}[\mathcal{G}_{m,n} \backslash X_1] \le \epsilon.$$

In addition, $\mathrm{SB}|_{X_2}$ is continuous by pasting lemma.

The rest of the proof is to apply Theorem C.10 for $X = X_2$, $\mathbf{G} = S_m \times S_n$, $\Phi = \mathrm{SB}$, and $\mathcal{F} = \mathcal{F}_{\text{2-FGNN}}$. We need to verify the four conditions in Theorem C.10. Condition (a) can be proved by similar arguments as in the proof of Lemma D.2 in [11]. Condition (b) follows directly from Theorem C.7 (a) and (c) and Theorem C.2. Condition (c) follows directly from Theorem C.7 (a) and Theorem C.3 (a). Condition (d) follows directly from Theorem C.7 (b) and Theorem C.3 (c). According to Theorem C.10, there exists some $F \in \mathcal{F}_{\text{2-FGNN}}$ such that

$$\sup_{G \in X_2} \|F(G) - \mathrm{SB}(G)\| \le \delta.$$

Therefore, one has

$$\mathbb{P}[\|F(G) - \mathrm{SB}(G)\| > \delta] \le \mathbb{P}[\mathcal{G}_{m,n} \backslash X_2] \le \epsilon,$$

which completes the proof. $\qquad \square$

# D  Extensions of the theoretical results

This section will explore some extensions of our theoretical results.

## D.1  Extension to other types of SB scores

The same analysis for Theorem 4.4 and Theorem 4.7 still works as long as the SB score is a function of $f_{\mathrm{LP}}^*(G, j, l_j, \hat{u}_j)$, $f_{\mathrm{LP}}^*(G, j, \hat{l}_j, u_j)$, and $f_{\mathrm{LP}}^*(G)$:

- We prove in Theorem A.3 that if two MILP-graphs are indistinguishable by the WL test, then they must be isomorphic and hence have identical SB scores (no matter how we define the SB scores). So Theorem 4.4 is still true.

- We prove in Theorem C.4 that if two MILP-graphs are indistinguishable by 2-FWL test, then they have the same value of $f_{\mathrm{LP}}^*(G, j, l_j, \hat{u}_j)$ (and $f_{\mathrm{LP}}^*(G, j, \hat{l}_j, u_j)$). Therefore, Theorem C.3 still holds if the SB score is a function of $f_{\mathrm{LP}}^*(G, j, l_j, \hat{u}_j)$, $f_{\mathrm{LP}}^*(G, j, \hat{l}_j, u_j)$, and $f_{\mathrm{LP}}^*(G)$, which implies that Theorem 4.7 is still true.

Therefore, Theorems 4.4 and 4.7 work for both linear product score functions in [14].

## D.2 Extension to varying MILP sizes

While Theorems 4.4 and 4.7 assume MILP sizes $m$ and $n$ are fixed, we now discuss extending these results to data distributions with variable $m$ and $n$.

First, our theoretical results can be directly extended to MILP datasets or distributions where $m$ and $n$ vary but remain bounded. Following Lemma 36 in [4], if a universal-approximation theorem applies to $\mathcal{G}_{m,n}$ for any fixed $m$ and $n$ (as shown in our work) and at least one GNN can distinguish graphs of different sizes, then the result holds across a disjoint union of finitely many $\mathcal{G}_{m,n}$.

If the distribution has unbounded $m$ or $n$, for any $\epsilon > 0$, one can always remove a portion of the tail to ensure boundedness in $m$ and $n$. In particular, there always exist large enough $m_0$ and $n_0$ such that $\mathbb{P}[m(G) \leq m_0] \geq 1 - \epsilon$ and $\mathbb{P}[n(G) \leq n_0] \geq 1 - \epsilon$. The key point is that for any $\epsilon > 0$, such $m_0$ and $n_0$ can always be found. Although these values may be large and dependent on $\epsilon$, they are still finite. This allows us to apply the results for the bounded-support case.

Note that the "tail removal" technique mentioned above comes from the fact that a probability distribution has a total mass of 1:

$$1 = \sum_{n=0}^{\infty} \mathbb{P}[n(G) = n] = \lim_{n_0 \to \infty} \sum_{n=0}^{n_0} \mathbb{P}[n(G) = n] = \lim_{n_0 \to \infty} \mathbb{P}[n(G) \leq n_0].$$

By the definition of a limit, this clearly implies that for any $\epsilon > 0$, there exists a sufficiently large $n_0$ such that $\mathbb{P}[n(G) \leq n_0] \geq 1 - \epsilon$. A similar argument applies to $m$.

# E   Details about numerical experiments

**Random MILP instances generation**   We generate 100 random MILP instances for the experiments in Section 5. We set $m = 6$ and $n = 20$, which means each MILP instance contains 6 constraints and 20 variables. The sampling schemes of problem parameters are described below.

- The bounds of linear constraints: $b_i \sim \mathcal{N}(0, 1)$.
- The coefficients of the objective function: $c_j \sim \mathcal{N}(0, 1)$.
- The non-zero elements in the coefficient matrix: $A_{ij} \sim \mathcal{N}(0, 1)$. The coefficient matrix $A$ contains 60 non-zero elements. The positions are sampled randomly.
- The lower and upper bounds of variables: $l_j, u_j \sim \mathcal{N}(0, 10^2)$. We swap their values if $l_j > u_j$ after sampling.
- The constraint types $\circ$ are randomly sampled. Each type ($\leq$, $=$ or $\geq$) occurs with equal probability.
- The variable types are randomly sampled. Each type (*continuous* or *integer*) occurs with equal probability.

**Implementation and training details**   We implement MP-GNN and 2-FGNN with Python 3.6 and TensorFlow 1.15.0 [1]. Our implementation is built by extending the MP-GNN implementation of [19] in `https://github.com/ds4dm/learn2branch`. The SB scores of randomly generated MILP instances are collected using SCIP [6].

For both GNNs, $p^0, q^0$ are parameterized as linear transformations followed by a non-linear activation function; $\{p^l, q^l, f^l, g^l\}_{l=1}^{L}$ are parameterized as 3-layer multi-layer perceptrons (MLPs) with respective learnable parameters; and the output mapping $r$ is parameterized as a 2-layer MLP. All layers map their input to a 1024-dimensional vector and use the ReLU activation function. Under these settings, MP-GNN contains 43.0 millions of learnable parameters and 2-FGNN contains 35.7 millions of parameters.

We adopt Adam [31] to optimize the learnable parameters during training with a learning rate of $10^{-5}$ for all networks. We decay the learning rate to $10^{-6}$ and $10^{-7}$ when the training error reaches $10^{-6}$ and $10^{-12}$ respectively to help with stabilizing the training process.

