# OpenReview forum: "Rethinking the Capacity of Graph Neural Networks for Branching Strategy"
_NeurIPS.cc/2024/Conference — NeurIPS 2024 poster_

### Official Review · Reviewer_VcHk · 2024-06-20

**Soundness:** 3
**Presentation:** 3
**Contribution:** 2
**Rating:** 5
**Confidence:** 4

**Summary:**

This submission analyzes the capacity of message-passing graph neural networks (MP-GNNs) in learning strong branching (SB) scores. A class of MILPs, so-called MP-tractable MILPs, are identified as a suitable class, whose SB scores could be accurately approximated by common MP-GNNs. For general MILPs, the authors also prove a similar result using more complex models, i.e., 2-FGNNs. Counterexamples are constructed to show that MP-GNNs cannot distinguish two different MP-intractable MILPs. Experiments on several small problems support the theoretical results.

**Strengths:**

This paper has three theoretical contributions:

- MP-GNNs are universal approximations to SB scores of MP-tractable MILPs.

- MP-GNNs can not approximate arbitrary MILPs accurately. Counterexamples are constructed.

- 2-FGNNs are universal approximations to SB scores of all MILPs.

The theoretical part is solid. I do not check every step but the skeleton of each proof looks quite reasonable. Considering the countless works on directly using various ML techniques to solve problems without fully thinking the compatibility between models and problems, such work is more valuable to me.

**Weaknesses:**

1. My first concern is about the MILP graph representation, which forms the basis of the proposed theories. Such representation is quite intuitive but relatively simple: the relationships between variables are not fully expressed. There could be multiple graphs with the same features, as well as the same local adjacent relationships, e.g., (4.1) and (4.2). I understand that there will be too many edges if one links any two variables appearing in the same constraint, and that 2-FGNN is a good alternative to capture such relationships without a over-complex graph structure. But I think the incapacity does not come from MP-GNNs but from the MILP graph representations.

2. My second concern comes from the numerical results.

- No experiments on MILP benchmarks. Testing on random instances with the same (and small) size is not enough to show that both MP-GNN and 2-FGNN are good approximations for all MP-tractable MILPs.

- The only test on (4.1) and (4.2) indeed shows that 2-FGNN outperforms MP-GNN. However, it is unclear that how much the solver can benefit from such improvement from $\sim 10^{-2}$ to $\sim 10^{-15}$ on SB score error, especially considering the extra cost on training $\sim 40000$ epochs. Providing the training time for both models and improved solving time using 2-FGNN will be more convincing.

- There is no information to show how common MP-intractable MILPs are. From my perspective, MP-intractability is mainly caused by symmetries between variables and constraints. I wonder how likely a real MILP problem has such symmetries.

**Questions:**

1. Have the authors considered adding more features to handle the incapacity? For instance, adding one more feature for each variable representing the number of variables associated with this variable by one or multiple constraints. In (4.1), this feature will be $8$ for all $8$ variables. In (4.2), however, this feature will be $3$ for $x_1,\dots, x_6$, while $2$ for $x_7$ and $x_8$. Then MP-GNN is probably capable of distinguishing these two MILPs.

2. Is root SB considered in all experiments? How much one can benefit from applying leaf SB? if one applies L2B in leaf nodes, I think local cuts will help break MP-intractability by involving more constraints (local cuts), or changing bounds of variables, etc..

3. The tested MILPs are quite small and there are just a few features, why using $3$ layers with $1024$ hidden features in each MLP? The models are large and very likely to overfit. Also, why there are fewer parameters in 2-FGNN comparing to MP-GNN as shown in lines 935 - 936?

**Limitations:**

See above for the limitations of this paper.

---

> ### Author Rebuttal · Authors · 2024-08-07
>
> Thank you very much for your valuable comments! Please find our responses below:
> * __[Weakness 1] MILP-graph representation:__ The bipartite graph representation is standard in the existing literature and has already been utilized in MILP-related learning tasks. The goal of our paper is to provide theoretical foundations and insights into this approach, specifically addressing what GNNs can and cannot represent in learning-to-branch (L2B) tasks.
>
>     To provide more context, we reference a sentence from our manuscript (lines 80-83 in Section 1): "To utilize a GNN on a MILP, one first conceptualizes the MILP as a graph and the GNN is then applied to that graph and returns a branching decision. This approach [14, 18] has gained prominence in not only L2B but various other MILP-related learning tasks [13, 16, 25, 29, 31, 35, 39, 42, 43, 47–50, 52, 55]."
>
> * __[Weakness 2] Benchmarks:__ Our work is mostly theoretical and aims to provide theoretical foundations for the learn-to-branch community. There have been numerous existing empirical works demonstrating the effectiveness of GNNs for branching strategies, as mentioned in the previously quoted sentence (lines 80-83 in Section 1 of our manuscript). These studies include real-world benchmarks such as MIPLIB 2017.
>
>     To further address the reviewer's concern, we __conducted additional experiments__ on larger instances. In this experiment, we trained an MP-GNN on 100 large-scale set covering problems with 1,000 variables and 2,000 constraints for each, which are generated following the practice in Gasse et al., (2019). The MP-GNN was successfully trained to achieve a training loss of $1.94\times 10^{-4}$, which was calculated as the average $\ell_2$ norm of errors on all training instances. We will add the discussion and the additional numerical results in the revised version.
>
> * __[Weakness 2] Training epochs:__ The purpose of the experiments in this paper is to directly validate our theoretical findings. In our experiment, we merely used basic end-to-end training without incorporating advanced techniques from the state-of-the-art literature. Therefore, the training performance shown in Figure 3 does not represent the upper limit of 2-FGNN and MP-GNN's training performance.
>
>     Improving the practical training efficiency of 2-FGNNs for MILP tasks is an interesting and significant future direction. One possible way to improve the training efficiency of high-order GNNs is to employ the sparsity as in Morris et al., (2020). We will add this discussion to our revised manuscript.
>
>     To further address the reviewer's concern, we have __conducted additional experiments__ on this issue. We adopted two additional training techniques that significantly decreased the training epochs needed. With an embedding size of 1,024, it now takes __980 epochs__ to reach $10^{-6}$ training loss and __1,128 epochs__ to reach $10^{-12}$ to fit the counter-examples as in Figure 3b. The two techniques introduced are:
>     * We let the two counter-examples come in stochastic order to break the symmetry of their gradients. Previously we used the full-dataset (batch size of 2) gradients for training.
>     * We used a linear layer to map the edge weights to 128-dimensional hidden embeddings. Previously we directly used the scalar edge weights as the embeddings.
>
> * __[Weakness 2] How common are MP-intractable MILPs:__ MP-intractability comes from the symmetry of MILPs. The symmetry of a MILP can be measured by the number of different colors in the output of the WL test. For example, (4.1) and (4.2) both admit two different colors, significantly less than the number of vertices in the corresponding MILP-graphs. In practice, even if there may not exist many examples with such strong symmetry, it is common to have some symmetry -- As reported by Chen et al., (2023), "for around 1/4 of the problems in MIPLIB 2017 (Gleixner et al., 2021), the number of colors generated by WL test is smaller than one half of the number of vertices".
>
> * __[Question 1] Adding more features:__ We appreciate your observation that adding more features is helpful for breaking the symmetry. The additional feature proposed by you (the number of associated variables by some constraint) can indeed distinguish (4.1) and (4.2). However, there exist other counter-examples with this additional feature. For example, one can add another constraint $\sum_{i=1}^8 x_i\geq 1$ to both (4.1) and (4.2). Then the additional feature becomes $8$ for all variables while the SB scores remain unchanged.
>
> * __[Question 2] Root SB vs Leaf SB:__ Yes, root SB is considered in our experiments. However, our theoretical findings can also be applied to leaf SB. This can be done by treating the leaf node as a new MILP and inputting this new graph into the GNNs.
>
>     We definitely agree with you that there might be less symmetry in leaf SB due to the local cuts.
>
> * __[Question 3] Number of parameters:__ We use a relatively large model size to demonstrate that the error/limitation of MP-GNN revealed in this paper is inherent; regardless of the number of parameters, even a large MP-GNN cannot fit the two toy instances shown in Figure 3(b).
>
>     To further address the reviewer's concern, we __conducted additional experiments__ on 2-FGNN training (using improved training techniques described as above) with varying embedding sizes ranging from 64 to 2,048. The results showed that a smaller embedding size, say 64, can still lead to a high accuracy $10^{-12}$.
>
>     | Embedding size | 64 | 128 | 256 | 512 | 1,024 | 2,048 |
>     |---|---|---|---|---|---|---|
>     | Epochs to reach $10^{-12}$ error | 18,762 | 7,474 | 4,412 | 2,484 | 1,128 | 1,174 |
>
>     Additionally, the reason that the number of parameters of 2-FGNN is comparable with (even a bit smaller than) MP-GNN is that only local updates are learnable and the aggregations are fixed.

---

> > ### Comment · Reviewer_VcHk · 2024-08-09
> >
> > Thanks for your response. I have no more questions.

---

### Official Review · Reviewer_xf7T · 2024-07-11

**Soundness:** 3
**Presentation:** 3
**Contribution:** 3
**Rating:** 5
**Confidence:** 4

**Summary:**

In this article the authors present new results regarding the expressivity of GNNs with respect to branching strategies in mixed integer linear programming (MILP). In particular, they explicit a class of MILPs (called MP-tractable) such that for each MILP in the class, there exists a MP-GNN that can ``mimic'' the strong branching heuristic in some senses, a popular and performant heuristic used in MILP solvers. More precisely, they show that the MP-GNN can compute a strong branching score (that the authors define), for each of the variable to be branched in the MILP. The authors also extend this result to more general MILPs, using some 2-Folklore GNN, a variant of the standard message passing GNNs. The main results are stated in Theorem 4.4 and Theorem 4.7, and hold in probability, given a distribution on the considered class of MILPs.

**Strengths:**

This paper contributes to lay some theoretical foundations of the use of GNNs for MILPs, and more generally combinatorial optimization. This is critical as the use of machine learning in optimization solvers is increasing.

The authors combine results in Approximation Theory (to approximate the SB function using GNNs) and known results about connections between color refinement and message passing GNNs.

Overall, this draft provides a serious study. I find the motivations and the current results interesting.
However, the paper in its current form has some issues, and I have have several concerns stated below.

**Weaknesses:**

- Concerning MP-tractability, I have a major concern that I state in the question list below (as I want to make sure I understood the result correctly).  I would be willing to revise my evaluation if the authors clarify this.

- The experiments are carried out on 100 instances, 6 constraints and 20 variables. The authors mention that one of the follow-ups is to conduct large-scale experiments, but a few thousand or say 10000 could already been more convincing (before large-scale ones).

- Some major statements are imprecise mathematically: Theorem 4.4. In which space does $G$ belong to: $G^{\text{MP}}$ or $G$?

- The notations and presentation of the proofs in the appendix is not very well written, making them difficult to follow and verify, despite their relative simplicity (in essence).

	Some examples:

	-  in Theorem A.3, the statement starts with $G \sim \bar{G}$, and a) states ``If $G \sim \bar{G}$ then [...]''
	-  in The statement of Theorem A.5, where you introduce $(\sigma_V, \sigma_W) * G$ without explaining the notation beforehand.
	-  Lemma C.8, one can group several substatements to simplify the presentation.

More generally, since the paper's main contributions are theoretical, the authors should reserve a section or subsection in the main paper to present sketches of proofs of their main results, that convey the key elements and ideas. Also, the authors should explain the main new ideas or techniques they introduced to obtain their results (or if they are essentially extensions or existing techniques, and so on), for example in comparison with the reference [11] that the authors cite.

**Questions:**

- Following first point in the weakness section: the authors mention that this gives a practical way of checking if one can compute the branching scores with a GNN. However, the very definition of MP-tractability implies that the partition obtained by color refinement is specific to the graph of the MILP. But to check that definition, one needs to verify this for every other graph input, and there are potentially exponentially many of them. This makes the verification computationally difficult, not polynomally as the authors claim.

- On the theoretical side, there is no discussion about the role of m and n in the results. In the current form of the statements, m and n are fixed, so the size of the graph inputs is fixed. Can the authors explain why this is a not a fundamental issue if one is interested in say, training a GNN that predicts SB scores across MILPs of different sizes?

**Limitations:**

The authors address in a few sentences the limitations and follow-up of their work in the conclusion.

---

> ### Author Rebuttal · Authors · 2024-08-07
>
> Thank you very much for your valuable comments! Please find our responses below:
>
> * __[Question 1, Weakness 1] Complexity of verifying MP-tractability:__ You are correct that verifying the MP-tractability of a MILP data set requires implementing the color refinement on each of the MILP-graphs in a data set.
>
>     Consider a set of MILPs, $\mathcal{D}$, containing $|\mathcal{D}|$ MILPs. To verify each of the MILPs in this set, one requires at most $\mathcal{O}(m+n)$ color refinement iterations according to Theorem 4.1. The complexity of each iteration is bounded by the number of edges in the graph (Shervashidze et al., 2011). In our context, it is bounded by the number of nonzeros in matrix $A$: $\text{nnz}(A)$. Therefore, the overall verification complexity is $\mathcal{O}(|\mathcal{D}|\cdot (m+n)\cdot \text{nnz}(A))$, which is linear in terms of $|\mathcal{D}|$, $(m+n)$, and $\text{nnz}(A)$. Note that $\text{nnz}(A) \leq m \times n$. This is why we say it is polynomial time.
>
>     In contrast, solving all the MILPs or even calculating all the SB scores in the given dataset $\mathcal{D}$ requires significantly higher complexity. To calculate the SB score of each MILP, one needs to solve at most $n$ LPs. We denote the complexity of solving each LP as $\text{CompLP}(m,n)$. Therefore, the overall complexity of calculating SB scores is $\mathcal{O}(|\mathcal{D}| \cdot n \cdot \text{CompLP}(m,n) )$. Note that, currently, there is still no strongly polynomial-time algorithm for LP, thus this complexity is significantly higher than that of verifying MP-tractability.
>
> * __[Question 2] The role of $m$ and $n$:__ You are correct that our current theory only considers fixed $m$ and $n$, which matches with existing literature on theories of GNNs for (MI)LPs (Chen et al., 2023a; 2023b). The theory can be extended directly to MILP dataset/distribution with varying but upper-bounded $m(G)$ and $n(G)$. Actually, the universal-approximation-type results can even be extended to unbounded sizes. This is because for any MILP distribution $\mathbb{P}$ and any $\epsilon>0$, there always exist large enough $m_0$ and $n_0$ such that $\mathbb{P}[m(G)\leq m_0]\geq 1-\epsilon$ and $\mathbb{P}[n(G)\leq n_0]\geq 1-\epsilon$. We will clarify this point in our revision.
>
> * __[Weakness 2] Large-scale numerical results:__ Our work is mostly theoretical and aims to provide theoretical foundations for the learn-to-branch community. There have been numerous existing empirical works demonstrating the effectiveness of GNNs for branching strategies. To support this point, we reference a sentence from our manuscript (lines 80-83 in Section 1): "To utilize a GNN on a MILP, one first conceptualizes the MILP as a graph and the GNN is then applied to that graph and returns a branching decision. This approach [14, 18] has gained prominence in not only L2B but various other MILP-related learning tasks [13, 16, 25, 29, 31, 35, 39, 42, 43, 47–50, 52, 55]."
>
>     To further address the reviewer's concern, __we conducted additional experiments__ on this matter. In this experiment, we trained an MP-GNN on 100 large-scale set covering problems with 1,000 variables and 2,000 constraints for each, which are generated following the practice in Gasse et al., (2019). The MP-GNN was successfully trained to achieve a training loss of $1.94\times 10^{-4}$, which was calculated as the average $\ell_2$ norm of errors on all training instances. We will add the discussion and the additional numerical results in the revised version.
>
> * __[Weakness 3] Statement in Theorem 4.4:__ While our theorem statement includes both notions $G_{m,n}$ and $G_{m,n}^{\textup{MP}}$, we would like to kindly emphasize that this statement is mathematically precise. First, we acknowledge that $G_{m,n}^{\textup{MP}} \subset G_{m,n}$. The probability distribution is built on the entire space $G_{m,n}$, not just the subspace $G_{m,n}^{\textup{MP}}$. We then assume that G belongs to the MP-tractable space $G \in G_{m,n}^{\textup{MP}}$ with probability one, which means $\mathbb{P}[G\in G_{m,n}\backslash G_{m,n}^{\textup{MP}}] = 0$.
>
> * __[Weakness 4] Presentation issues:__ In our revision, we will make sure to reserve a subsection in the main paper to clearly sketch the proof and explain the main ideas/techniques used in the proof. We will also clarify some notation including:
>     * There is a typo in the statement of Theorem A.3. It should be "For any $G,\bar{G}\in G_{m,n}^{\textup{MP}}$ ..."
>     * In Theorem A.5, $(\sigma_V,\sigma_W)\ast G$ is the MILP-graph obtained from $G$ by reordering vertices with permutations $\sigma_V$ and $\sigma_W$.
>     * Lemma C.8: We will group certain substatements for clarity: (a) and (c) both represent the equivalence between vertices within a single graph, so they can be merged. However, (b) represents the equivalence between vertices of two different graphs and cannot be trivially merged with (a). Similarly, (d) and (f) can be merged as they both represent the equivalence within a single graph.
>
> __References:__ (Shervashidze et al., 2011) Nino Shervashidze, Pascal Schweitzer, Erik Jan Van Leeuwen, Kurt Mehlhorn, and Karsten M. Borgwardt. Weisfeiler-lehman graph kernels. Journal of Machine Learning Research 12, no. 9, 2011.
>
> (Chen et al., 2023a) Ziang Chen, Jialin Liu, Xinshang Wang, Jianfeng Lu, and Wotao Yin. On representing linear programs by graph neural networks. In The Eleventh International Conference on Learning Representations, 2023.
>
> (Chen et al., 2023b) Ziang Chen, Jialin Liu, Xinshang Wang, Jianfeng Lu, and Wotao Yin. On representing mixed-integer linear programs by graph neural networks. In The Eleventh International Conference on Learning Representations, 2023.
>
> (Gasse et al., 2019) Maxime Gasse, Didier Chételat, Nicola Ferroni, Laurent Charlin, and Andrea Lodi. Exact combinatorial optimization with graph convolutional neural networks. Advances in Neural Information Processing Systems, 32, 2019.

---

> > ### Comment · Reviewer_xf7T · 2024-08-09
> >
> > Thank you for your detailed answers. I would like to follow up with additional question(s) to the authors about the main theoretical contribution.
> >
> > Q1: Verifying MP-tractability can indeed be performed in polynomial time with respect to the dataset's size (and other constants / parameters). So given a set of instances, one can verify in a reasonable time if they are MP-tractable. However, in the statement of Theorem 4.4, the GNN's size potentially depends on the level of error and probability. The size of the GNN to insure one has low error and high probability may be exponential (w.r.t to $\epsilon$ and $\delta$). Also, the $\epsilon$ and $\delta$ have to grow with the size of the instances to make sure one is close enough to the true branching score. Therefore, the poly-time MP-tractable testing advertised by the authors could be overtaken by the GNN's size in this case (let alone the complexity to actually find or train this GNN). Can the authors comment on this?
> > In that regard, a clear presentation of the nature of the result in the contributions section (or elsewhere in the article), which is (as far as I understood), existential and not constructive, could be helpful.
> >
> > Q2: Do you mean that the statement would hold for any distribution (say with total weight 1)?
> > Would it help to consider bounded-support distributions (in m and n), if this allows to remove technical difficulties? (and is probably relevant to most applications).

---

> > > ### Author Response · Authors · 2024-08-11
> > > **Reply to Reviewer xf7T**
> > >
> > > Thanks for your additional questions! Here are our answers:
> > >
> > > **Reply to Q1.** We completely agree with you that our main results focus on the existence of GNNs representing the SB score, and we have not yet established the complexity of such GNNs. In our revision, we will clearly comment that verifying MP-tractability is polynomial in complexity, but the size and running time of GNNs currently have no theoretical guarantees.
> > >
> > > However, we would like to kindly emphasize that this does not hurt our main contribution. The main purpose of this paper is to address a theoretical question: **whether GNNs can or cannot universally represent the SB score** for all MILPs, especially given the numerous empirical studies on this topic. We clearly answer this question with Theorem 4.4, Corollary 4.6, and Theorem 4.7—when an MILP exhibits strong symmetry (such as in equations (4.1) and (4.2)), MP-GNNs are not suitable, and other GNN structures need to be explored. This conclusion is inherent, regardless of how large MP-GNNs might be, and we believe it provides useful insights to the learning-to-branch community.
> > >
> > > As for the complexity of these GNNs (whether polynomial or exponential), this relates to a different question: **how well GNNs can represent the SB score.** We believe this is an interesting direction for future research.
> > >
> > > To further address the reviewer's concern, we will include the following in our revision:
> > >
> > > > At the end of Section 4.3, we will add: While verifying MP-tractability is polynomial in complexity, this does not imply that calculating the SB score is polynomial, as the complexity of GNNs is not guaranteed. Our Theorems 4.4 and 4.7 address existence, not complexity. In other words, this paper answers the question of whether GNNs can represent the SB score. To explore how well GNNs can represent SB, further investigation is needed.
> > >
> > > We chose to include the comment at the end of Section 4.3 because it involves concepts like MP-tractability and Theorems 4.4 and 4.7. Positioning the comment after these concepts and theorems are introduced ensures a smoother flow for the readers.
> > >
> > >
> > > **Reply to Q2.** Yes, the results hold for any Borel regular probability distribution. We would like to provide more details on how to extend our results to a data distribution with varying sizes $m,n$.
> > >
> > > According to Lemma 36 in (Azizian and Lelarge, 2021), if one can obtain a universal-approximation-type theorem on $G_{m,n}$ for any **fixed** $m$ and $n$ (as we addressed in our manuscript), and if graphs with different sizes can be distinguished by at least one GNN (this is straightforward), then the result can be extended directly to the disjoint union of **finitely many** $G_{m,n}$. This addresses the case of "bounded-support distributions (in $m$ and $n$)" mentioned by you.
> > >
> > > If the distribution is not bounded-support in $m$ and $n$, for any $\epsilon>0$, one can always remove a portion of the tail to ensure boundedness in $m$ and $n$. That is what we mentioned in the rebuttal: there always exist large enough $m_0$ and $n_0$ such that $\mathbb{P}[m(G)\leq m_0]\geq 1-\epsilon$ and $\mathbb{P}[n(G)\leq n_0]\geq 1-\epsilon$. The key point is that for any $\epsilon>0$, such $m_0$ and $n_0$ can always be found. Although these values may be large and dependent on $\epsilon$, they are still finite. This allows us to apply the results for the bounded-support case.
> > >
> > > Note that the "tail removal" technique mentioned above comes from the fact that a probability distribution has a total mass of 1:
> > > $$1 = \sum_{n=0}^{\infty}\mathbb{P}(n(G) = n) = \lim_{n_0 \to \infty} \sum_{n=0}^{n_0}\mathbb{P}(n(G) = n) = \lim_{n_0 \to \infty} \mathbb{P}(n(G)\leq n_0)$$
> > > By the definition of a limit, this clearly implies that for any $\epsilon> 0$, there exists a sufficiently large $n_0$ such that $\mathbb{P}[n(G)\leq n_0]\geq 1-\epsilon$. A similar argument applies to $m$.
> > >
> > > Please let us know if you have further comments or questions!
> > >
> > > **(References).** (Azizian and Lelarge, 2021) Waiss Azizian and Marc Lelarge. Expressive power of invariant and equivariant graph neural networks. In International Conference on Learning Representations, 2021.

---

> > > > ### Comment · Reviewer_xf7T · 2024-08-13
> > > >
> > > > Thanks for the detailed answers. I am satisfied with the authors' answers and raise my score to 5.
> > > > I am convinced that the core of the results are interesting. However, a significant improvement the presentation of the contributions (including mentioning limitations as explained in Q1 above), and the proofs (including some more developments to present their essence), could have improved my overall score further. Given the fact that this would be substantial modifications, I decided to raise to 5.

---

> ### Author Response · Authors · 2024-08-13
>
> Dear Reviewer xf7T,
>
> Thank you very much for increasing the score! We appreciate all your valuable comments and will modify the presentation and proof flows accordingly in our revision.

---

### Official Review · Reviewer_1NEU · 2024-07-13

**Soundness:** 3
**Presentation:** 3
**Contribution:** 3
**Rating:** 7
**Confidence:** 3

**Summary:**

This paper concerns the expressive power of GNN in the context of approximating Strong Branching scores in learning to branch. In this paper, the authors proposed the notion of "MP-tractable" Mixed Integer Linear Programs (MILPs) and analytically proved that all MP-tractable MILPs are distinguishable by message passing GNNs - a structure frequently used in state-of-the-art models in learning to branch. The authors then demonstrated through example that the same may not hold non-MP-tractable MILPs, and further provided analytical that all MILPs, regardless of MP-tractability, can be distinguished by 2-FGNNs. A small scale numerical experiment is included as a prove of concept.

**Strengths:**

1. The paper is rigorously written and easy to follow, claims are accompanied with theoretical justifications
2. Examines the limitations of a commonly used tactic in Learning to Branch, with proposed necessary condition for being GNN-distinguishable weaker than previous work.

**Weaknesses:**

1. The proofs seems to be dependent on the assumption that the LP solution is one of minimum L2 norm, which somewhat limits its implication in real-life scenarios as such solution is often not selected by LP solving algorithms like simplex

**Questions:**

1. This paper focused on product-type strong branching scores in their analysis. While it is popularly used as the default scoring scheme in SCIP, there are other strong branching rules that exists as well (see [1]). How does your result generalizes to different strong branching scores, and what properties does the strong branching scheme need for your results to be applied?

[1] Dey, S.S., Dubey, Y., Molinaro, M. et al. A theoretical and computational analysis of full strong-branching. Math. Program. 205, 303–336 (2024). https://doi.org/10.1007/s10107-023-01977-x

**Limitations:**

The authors have discussed the limitations of the work. The large-scale experiments are still a problem in existing studies.

---

> ### Author Rebuttal · Authors · 2024-08-07
>
> Thank you very much for your encouraging comments! Please find our responses below:
> * __LP solution with the smallest $\ell_2$-norm:__ One reason we choose LP solution with the smallest $\ell_2$-norm rather than an arbitrary one is to make sure that the LP solution, as well as its resulting SB score, is uniquely defined. Another reason for this assumption is to keep the theoretical results clean and simple while providing sufficient insight -- When the dataset has strong symmetry, it is not suitable to use MP-GNNs and one can explore other GNN structures.
>
>     Additionally, we will mention that in some cases, the assumption (LP solution with the smallest $\ell_2$-norm) can be directly applied: (1) The LP relaxation admits a unique optimal solution. (2) The solution selected by LP algorithms and the solution with the smallest $\ell_2$-norm have the same output with the floor and ceiling functions.
>
> * __Other types of strong branching scores:__ The same analysis for Theorem 4.4 and Theorem 4.7 still works as long as the SB score is a function of $f_{\textup{LP}}^*(G,j,l_j,\hat{u_j})$, $f_{\textup{LP}}^*(G,j,\hat{l_j},u_j)$, and $f_{\textup{LP}}^*(G)$:
>
>     * We prove in Theorem A.3 that if two MILP-graphs are indistinguishable by the WL test, then they must be isomorphic and hence have identical SB scores (no matter how we define the SB scores). So Theorem 4.4 is still true.
>     * We prove in Theorem C.4 that if two MILP-graphs are indistinguishable by 2-FWL test, then they have the same value of $f_{\textup{LP}}^*(G,j,l_j,\hat{u_j})$ (and $f_{\textup{LP}}^*(G,j,\hat{l_j},u_j)$). Therefore, Theorem C.3 still holds if the SB score is a function of $f_{\textup{LP}}^*(G,j,l_j,\hat{u_j})$, $f_{\textup{LP}}^*(G,j,\hat{l_j},u_j)$, and $f_{\textup{LP}}^*(G)$, which implies that Theorem 4.7 is still true.
>
>     In particular, Theorem 4.4 and Theorem 4.7 work for both linear score function and product score function in Dey et al., (2024). Additionally, it can be easily verified (with almost the same calculation as in Appendix B) that the counter-examples (4.1) and (4.2) also work for the linear score function. We will clarify this point in our revision.
>
> __References:__ (Dey et al., 2024) Santanu S. Dey, Yatharth Dubey, Marco Molinaro, and Prachi Shah. "A theoretical and computational analysis of full strong-branching." Mathematical Programming 205(1):303-336, 2024.

---

> > ### Comment · Reviewer_1NEU · 2024-08-12
> >
> > My only concern was on practical impact/generalizability of result, and it has been adequately addressed in the rebuttal. I have no further comment.

---

### Official Review · Reviewer_uxqS · 2024-07-13

**Soundness:** 3
**Presentation:** 4
**Contribution:** 3
**Rating:** 8
**Confidence:** 2

**Summary:**

This paper provides a new lens through which the capacity of GNNs can be analyzed---branching strategy. The correspondence between strong branching and GNNs is established and the expressiveness of GNNs are discussed in terms of whether universally approximating strong branching is possible.

**Strengths:**

- the idea is novel and sound
- the presentation of the paper is excellent and easy to follow
- the notion of going beyond WL test to study the expressiveness of GNNs are likely to encourage and guide the design of more expressive GNNs

**Weaknesses:**

- only one experiment (in Figure 2b) is employed to showcase the difference in the expressive power between MILP and MP-GNN, and the difference only shows after 30,000 epochs. How would this practically affect the real-life experiments?

**Questions:**

- what would the behavior in Figure 3b change if the number of parameters were varied?

**Limitations:**

The limitations have been sufficiently addressed in the paper.

---

> ### Author Rebuttal · Authors · 2024-08-07
>
> Thank you very much for your encouraging comments! Please find our responses below:
> * __Practical impact:__ The purpose of the experiment in Figure 3b is to illustrate the difference between 2-FGNN and MP-GNN for MILP problems with symmetry (beyond the MP-tractable class). Our paper is mostly theoretical and the current experiment is only for validating the theoretical findings. While we have not yet achieved highly efficient training of 2-FGNN in practice (which will be a future research direction), we believe that our results provide theoretical insights and offer guidance to the learn-to-branch community -- When the dataset has strong symmetry, it is not suitable to use MF-GNNs and one can explore other GNN structures.
>
>     It's important to note that in our experiment, we merely used basic end-to-end training without incorporating advanced techniques from the state-of-the-art literature. Therefore, the training performance shown in Figure 3 does not represent the upper limit of 2-FGNN and MP-GNN's training performance.
>
>     To further address the reviewer's concern, __we conducted additional experiments__ on this matter. We adopted two additional training techniques that significantly decreased the training epochs needed. With an embedding size of 1,024, it now takes __980 epochs__ to reach $10^{-6}$ training loss and __1,128 epochs__ to reach $10^{-12}$ to fit the counter-examples as in Figure 3b. The two techniques introduced are:
>     * We let the two counter-examples come in stochastic order to break the symmetry of their gradients. Previously we used the full-dataset (batch size of 2) gradients for training.
>     * We used a linear layer to map the edge weights to 128-dimensional hidden embeddings. Previously we directly used the scalar edge weights as the embeddings.
>
> * __Number of parameters:__ In Figure 3b, the behavior of MP-GNN will not change no matter how many parameters we use, which is guaranteed by Theorem 4.5. This error is inherently related to the symmetry structure of MP-intractable MILPs and cannot be eliminated by increasing the number of parameters. In contrast, the loss of 2-FGNN can be arbitrarily close to $0$ if there are sufficiently many parameters, which is guaranteed by Theorem 4.7 and validated in our numerical experiment.
>
>     To further address the reviewer's concern, __we conducted additional experiments__ on 2-FGNN training (using improved training techniques described above) with varying embedding sizes ranging from 64 to 2,048. We observed that all models achieved near-zero errors but took different numbers of epochs as shown in the table below. The results showed that overall an increased embedding size enlarges the model capacity to fit the counter-examples, which saturates when the embedding size is over 1,024. This may be attributed to the increased training difficulty because the embedding size becomes too large.
>
>     | Embedding size | 64 | 128 | 256 | 512 | 1,024 | 2,048 |
>     |-----------------------|-----|-------|------|-------|---------|---------|
>     | Epochs to reach $10^{-6}$ error | 16,570 | 5,414 | 2,736 | 1,442 | 980 | 1,126 |
>     | Epochs to reach $10^{-12}$ error | 18,762 | 7,474 | 4,412 | 2,484 | 1,128 | 1,174 |

---

> > ### Comment · Reviewer_uxqS · 2024-08-08
> >
> > Many thanks for your rebuttal. My concerns are all sufficiently addressed.

---

### Official Review · Reviewer_BDoF · 2024-07-17

**Soundness:** 4
**Presentation:** 3
**Contribution:** 3
**Rating:** 8
**Confidence:** 5

**Summary:**

The paper investigates the effectiveness of GNNs in approximating strong branching (SB) strategies in mixed-integer linear programming (MILP) problems. SB is a heuristic used in the branch-and-bound algorithm to choose branching variables. SB is among the best-performed heuristics in branch-and-bound algorithm, but is extremely computationally expensive.

### Key Contributions:
1. The paper defines a class of MILPs called "MP-tractable" for which message-passing GNNs (MP-GNNs) can approximate SB scores accurately. It establishes a universal approximation theorem for MP-GNNs within this class.
2. The study shows that MP-GNNs cannot represent SB scores for MILPs beyond the MP-tractable class. This is demonstrated through counter-examples where MP-GNNs fail to distinguish different MILP instances with distinct SB scores.
3. The paper proposes second-order folklore GNNs (2-FGNNs), which overcome the limitations of MP-GNNs and can universally approximate SB scores across any MILP distribution. Small-scale numerical experiments are performed to support the claim.

**Strengths:**

SB is a very important heuristic in branch-and-bound. It is strongly desired to develop a computationally efficient method to approximate SB score, so as to accelerate the solving of MILP. This paper proves that traditional MP-GNN can not universally approximate SB scores, and proposes a new GNN framework that can universally approximate SB scores. This theoretical insight is of great importance to the L2O community.

**Weaknesses:**

1. The paper provides a nice theoretical justification for why MP-GNN can not handle the intrinsic symmetry in SB scores. However, it is not clear whether such symmetry causes issues in practice. The symmetric case such as the counter-example in this paper is very rare in common MILP problems.

2. Even for small-scale data, 2-FGNN needs more than 30,000 epochs to fit the MP-intractable data. The lack of learning efficiency shows that 2-FGNN is far from practical.

**Questions:**

The paper is generally well-written. Besides the issues in the section of ``Weakness", I don't have further questions so far.

**Limitations:**

Yes.

---

> ### Author Rebuttal · Authors · 2024-08-07
>
> Thank you very much for your encouraging comments! Please find our responses below:
> * __Symmetry of MILPs in practice:__ The symmetry of a MILP problem can be measured by the number of different colors in the output of WL test. For example, (4.1) and (4.2) both admit two different colors, significantly less than the number of vertices in the corresponding MILP-graphs. This phenomenon frequently occurs in practical MILP datasets such as MIPLIB 2017. As noted in Chen et al., (2023), "for around 1/4 of the problems in MIPLIB 2017 (Gleixner et al., 2021), the number of colors generated by WL test is smaller than one half of the number of vertices,"  indicating that approximately 1/4 of the problems exhibit symmetry. We will add this discussion to our revised manuscript.
> * __Training efficiency:__ We agree that our work is primarily theoretical, and our experiments were designed to directly validate our theoretical findings. In our experiment, we merely used basic end-to-end training without incorporating advanced techniques from the state-of-the-art literature. Therefore, the training performance shown in Figure 3 does not represent the upper limit of 2-FGNN and MP-GNN's training performance.
>
>     Improving the practical training efficiency of 2-FGNNs for MILP tasks is an interesting and significant future direction. One possible way to improve the training efficiency of high-order GNNs is to employ the sparsity as in Morris et al., (2020). We will add this discussion to our revised manuscript.
>
>     To further address the reviewer's concern, __we have conducted additional experiments__ on this issue. We adopted two additional training techniques that significantly decreased the training epochs needed. With an embedding size of 1,024, it now takes __980 epochs__ to reach $10^{-6}$ training loss and __1,128 epochs__ to reach $10^{-12}$ to fit the counter-examples as in Figure 3b. The two techniques introduced are:
>     * We let the two counter-examples come in stochastic order to break the symmetry of their gradients. Previously we used the full-dataset (batch size of 2) gradients for training.
>     * We used a linear layer to map the edge weights to 128-dimensional hidden embeddings. Previously we directly used the scalar edge weights as the embeddings.
>
> __References:__ (Chen et al., 2023) Ziang Chen, Jialin Liu, Xinshang Wang, Jianfeng Lu, and Wotao Yin. On representing mixed-integer linear programs by graph neural networks. In The Eleventh International Conference on Learning Representations, 2023.
>
> (Morris et al., 2020) Christopher Morris, Gaurav Rattan, and Petra Mutzel. Weisfeiler and Leman go sparse: Towards scalable higher-order graph embeddings. Advances in Neural Information Processing Systems, 33:21824–21840, 2020.

---

> > ### Comment · Reviewer_BDoF · 2024-08-13
> >
> > Thanks for the response.
> >
> > **Symmetry of MILPs in practice**: My original meaning is "MP-intractable case is rare", not "symmetric case is rare". As long as the MILP is MP-tractable, GNN can handle no matter how much symmetry (in terms of the number of different colors output by the WL-test) exists.
> >
> > **Training efficiency**: I appreciate the authors' effort in the additional experiments. I agree that more advanced training techniques or new network architecture in the future can resolve this problem.

---

> ### Author Response · Authors · 2024-08-14
>
> __MP-intractability vs Symmetry:__ Thank you very much for your response and clarification! The density of MP-intractable MILPs varies across datasets and needs to be verified numerically depending on the dataset. Fortunately, verifying MP-tractability in practice can be done efficiently.
>
> To verify MP-intractability of a MILP, one requires at most $\mathcal{O}(m+n)$ color refinement iterations according to Theorem 4.1. The complexity of each iteration is bounded by the number of edges in the graph (Shervashidze et al., 2011). In our context, it is bounded by the number of nonzeros in matrix $A$: $\text{nnz}(A)$. Therefore, the overall complexity of the color refinement algorithm is $\mathcal{O}((m+n)\cdot \text{nnz}(A))$, which is linear in terms of $(m+n)$ and $\text{nnz}(A)$.
>
> In contrast, solving an MILP or even calculating its all the SB scores requires significantly higher complexity. To calculate the SB score of each MILP, one needs to solve at most $n$ LPs. We denote the complexity of solving each LP as $\text{CompLP}(m,n)$. Therefore, the overall complexity of calculating SB scores is $\mathcal{O}( n \cdot \text{CompLP}(m,n) )$. Note that, currently, there is still no strongly polynomial-time algorithm for LP, thus this complexity is significantly higher than that of verifying MP-tractability.
>
> These discussions will be included in our revision. We hope these discussions address your concern.
>
> __References:__ (Shervashidze et al., 2011) Nino Shervashidze, Pascal Schweitzer, Erik Jan Van Leeuwen, Kurt Mehlhorn, and Karsten M. Borgwardt. Weisfeiler-lehman graph kernels. Journal of Machine Learning Research 12, no. 9, 2011.

---

### Decision · Program_Chairs · 2024-09-25

**Decision:**

Accept (poster)

**Comment:**

After the rebuttal, all five reviewers agreed that this paper is worth being accepted, mainly for its important theoretical contribution regarding the expressive power of GNNs for strong branching, a strategy used in (the theory and practice) of mixed-integer programming. "Learning to branch" is a big deal in the MIP world, and this paper provides important fundamental insights into what is generally possible.

Some doubts regarding the presentation remain, and even though everyone is ok with acceptance in the end, I urge the authors to take the reviewer comments very serious when preparing the final version.